# AN END-TO-END MODEL FOR LOGITS BASED LARGE LANGUAGE MODELS WATERMARKING

## ABSTRACT

The rise of large language models (LLMs) has increased concerns over source tracing and copyright protection for AI-generated content (AIGC), highlighting the need for advanced detection technologies. Passive detection methods usually face high false positives, while active watermarking techniques using logits or sampling manipulation offer more effective protection. Existing LLM watermarking methods, though effective on unaltered content, suffer significant performance drops when the text is modified and could introduce biases that degrade LLM performance in downstream tasks. These methods fail to achieve an optimal tradeoff between text quality and robustness, particularly due to the lack of end-to-end optimization of the encoder and decoder. In this paper, we introduce the first end-to-end logits perturbation method for watermarking LLM-generated text. By jointly optimizing the encoder and decoder, our approach achieves a better balance between quality and robustness. To address non-differentiable operations in the end-to-end training pipeline, we introduce an online prompting technique that leverages the on-the-fly LLM as a differentiable surrogate. Our method demonstrates superior detection robustness, consistently outperforming state-of-the-art (SOTA) methods by 1.2%, 4.0%, and 5.5% across 3 LLMs, averaged over 6 types of text distortions. Simultaneously, our approach achieves exceptional text quality, as evidenced by reduced text perplexity and improved performance in the downstream tasks with a margin of 19.2% and 3.03%. Our method can be easily generalized to different LLMs. The code is available in supplementary material.

## 1 INTRODUCTION

Large Language Models (LLMs) like ChatGPT (Achiam et al., 2023), Llama (Touvron et al., 2023; Dubey et al., 2024), and OPT (Zhang et al., 2022) have greatly improved the quality of AI-generated content (AIGC), broadening their applications in fields such as translation (Hendy et al., 2023), content creation (Ni et al., 2023), etc. However, such a rapid expansion has also raised concerns about copyright infringement, academic dishonesty, and unethical practices (Augenstein et al., 2024). These issues highlight the urgent need for reliable methods to distinguish between human-written text and LLM-generated content, ensuring digital integrity and combating misinformation (Barrett et al., 2023).

Numerous approaches have been proposed to address LLM ethical concerns by detecting LLM-generated content. Passive detection methods focus on identifying unique properties of generated text, often through training binary classifiers (Bakhtin et al., 2019; Jawahar et al., 2020) or statistical techniques like DetectGPT (Mitchell et al., 2023), which compares the log probability value of a sentence with its perturbed version. However, as LLMs improve and the gap between generated and human-written text narrows, the effectiveness of these methods declines dramatically. In contrast, active detection methods, like embedding watermarks in generated text, are proving to be more robust alternatives. LLM watermarking methods fall into two main categories: logits-based and sampling-based. Specifically, logits-based methods (Kirchenbauer et al., 2023; Liu et al., 2024b; Huo et al., 2024) randomly divide the vocabulary into "green" and "red" lists by hashing preceding tokens as the seed. Then, perturbations are introduced to the logits that favor green list tokens in the generated text, and the proportion of the "green" tokens is used to distinguish whether a text is LLM-generated. Sampling-based methods (Kuditipudi et al., 2024; Christ et al., 2024) rely on random bitstream to guide token sampling, creating detectable correlations in the text. Despite

advancements, current watermarking schemes experience significant performance degradation with even slight text modifications. Additionally, existing algorithms introduce logit biases or guide sampling through random bitstream, which would result in semantic differences between watermarked and non-watermarked content, negatively impacting LLM performance on downstream tasks, and limiting the practicality of these watermarking techniques. Among logits-based methods, there is a growing trend to replace the hashing schemes with trainable networks for generating logit perturbations (Liu et al., 2024b; Huo et al., 2024), and to substitute statistical decoding with trainable decoders (Liu et al., 2024a), leveraging the flexibility of neural network training to improve performance. However, these existing approaches still fail to achieve an optimal trade-off between text quality and robustness, primarily due to the separate training of the encoder and decoder rather than a joint, end-to-end optimization. With this observation, we introduce the first logits-based end-to-end model, where lightweight encoder and decoder networks are jointly optimized to enhance both detecting robustness and text quality. Note that building such an end-to-end system is highly non-trivial because many involved modules, such as the complex text modification and the semantic loss computation, are inherently non-differentiable. To resolve these challenges brought by the non-differentiability, we introduce a novel online prompting technique that utilizes the on-the-fly LLM as a differentiable surrogate. This approach enables the model to effectively handle the above-mentioned non-differentiable operations. Our framework is LLM-agnostic, allowing any LLM to be used during training, and once trained, the model can be applied to other LLMs without retraining.

Our key contributions are as follows:

- For the first time, we present a logits-based end-to-end model for LLM watermarking, improving detection robustness and text quality through encoder-decoder joint optimization.
- We introduce an online prompting technique that transforms non-differentiable operations, such as semantic loss calculation and advanced online text modification, into differentiable operations by dynamically prompting the on-the-fly LLM. This technique enables seamless end-to-end training without external models.
- Extensive experiments show that our method surpasses SOTA methods by 1.2%, 4.0%, and 5.5% across 3 LLMs and 6 distortion types. It also achieves superior text quality, reducing perplexity and improving downstream task performance by 19.2% and 3.03%. Notably, our method can be generalized across LLMs without additional training.

The paper is organized as follows: Sec. 2 reviews related works on LLM watermarking. Sec. 3 details our proposed method. Sec. 4 presents experimental results demonstrating the superior performance of our method. Finally, Sec. 5 concludes the paper.

## 2    RELATED WORKS ON LLM WATERMARKING

Recent works on LLM watermarking can be classified into two main categories: logits-based and sampling-based methods. Logits-based methods, as presented in KGW (Kirchenbauer et al., 2023), split the vocabulary into "green" and "red" lists by hashing preceding tokens and biasing green list logits to favor their selection, using the proportion of green tokens to detect watermarks. Building on KGW, several methods aim to improve the robustness, quality, or unforgeability of the watermarked text. KGW-R (Kirchenbauer et al., 2024) explores different hashing schemes, while Unigram (Zhao et al., 2024) uses a fixed red-green separation to enhance editing resistance. SWEET (Lee et al., 2023) selectively modifies logits at high-entropy tokens to improve quality in low-entropy scenarios such as the code generation. UPV (Liu et al., 2024a) employs an encoder network to split lists and a detector network for classification, enabling public detection. SIR (Liu et al., 2024b) trains an encoder to apply context-aware biases for better robustness, and TSW (Huo et al., 2024) uses two networks to adaptively adjust watermark strength and list-splitting ratio for a better balance between detectability and text quality. Sampling-based methods, such as EXP (Kuditipudi et al., 2024), use a pseudo-random bitstream to guide token selection through the exponential sampling rule. This process produces watermarked text that aligns with the bit sequence and makes detection accurate. EXP-Edit (Christ et al., 2024) builds on the EXP method by introducing edit distance to measure sequence alignment, enhancing robustness against tampering. Nevertheless, as will be shown in the experiments, current LLM watermarking methods suffer from significant drops in detection performance when the text is modified and can introduce biases that impair LLM performance in common use cases, such as the translation and the code generation.

## 3   METHOD

In this section, we detail our LLM watermarking solution, which improves the balance between robustness and text quality, compared to existing methods. We begin by outlining LLM and logits-based watermarking basics, followed by the structure and objectives of our model and the online prompting technique. Finally, we introduce a plug-and-play converter for cross-LLM inference.

First, let us give a brief overview of the LLM workflow and the logits-based method. Given a prompt $\mathbf{X}_{\text{prompt}} = [\mathbf{x}_1, \ldots, \mathbf{x}_k]$, an LLM $M$ generates tokens in an autoregressive manner. At each time step $t$, $M$ produces a probability distribution for the next token $p_M(\mathbf{x}_t | \mathbf{x}_1, \ldots, \mathbf{x}_{t-1})$ over vocabulary $\mathcal{V} = \{\mathbf{t}_1, \ldots, \mathbf{t}_{|\mathcal{V}|}\}$, then the token $\mathbf{x}_t$ is sampled from $p_M(\mathbf{x}_t)$ . In this process, logits $\mathbf{l}^{(t)} = [l_1^{(t)}, \ldots, l_{|\mathcal{V}|}^{(t)}] \in \mathbb{R}^{|\mathcal{V}|}$ refer to the unnormalized output by $M$ before converting into probability

$$p_M(\mathbf{x}_t = \mathbf{t}_k | \mathbf{x}_1, \ldots, \mathbf{x}_{t-1}) = \frac{\exp(l_k^{(t)})}{\sum_{i=1}^{|\mathcal{V}|} \exp(l_i^{(t)})}. \tag{1}$$

Once the stop criteria are satisfied, the user receives a generated response $\mathbf{X}_{\text{nwm}}$. To embed a watermark into the generated text, logits-based methods introduce a watermark logits $\mathbf{l}_W^{(t)}$ at each generation step with a strength $\delta$, resulting in the perturbed logits: $\hat{\mathbf{l}}^{(t)} = \mathbf{l}^{(t)} + \delta \cdot \mathbf{l}_W^{(t)}$. By adjusting the logits to favor tokens in the green list, which is determined by hashing preceding tokens, the method increases the probabilities of sampling green-list tokens. Consequently, the watermarked text $\mathbf{X}_{\text{wm}}$ contains a higher proportion of these favored tokens, creating a detectable statistical cue that is not present in human-written content.

We are now ready to present the details of our end-to-end watermark method.

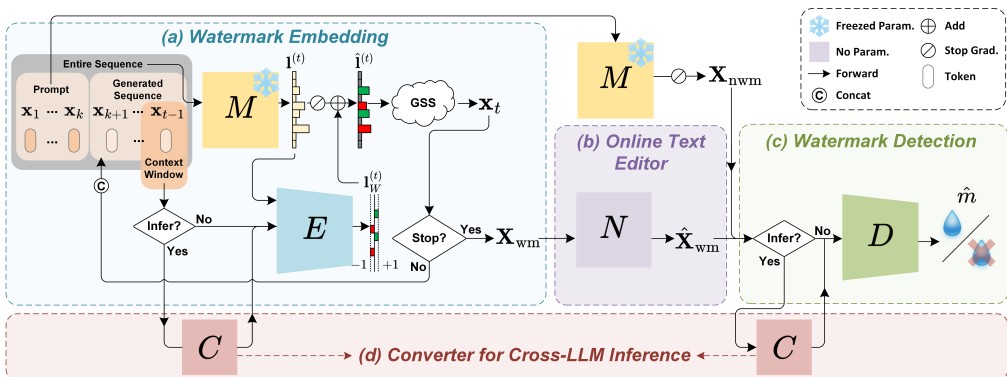

Figure 1: Overview of our end-to-end model, consisting of (a) watermarking via logits perturbation with encoder $E$; (b) simulating user edits using the online text editor $N$; and (c) detecting watermarked content through decoder $D$. The entire model is trained end-to-end to optimize both the quality and detection accuracy of the watermarked content. In the inference phase, (d) a converter is deployed for cross-LLM adaption. GSS is the abbreviation of the Gumbel-Softmax sampling.

### 3.1   MODEL OVERVIEW

The architecture of our proposed method is depicted in Fig. 1. At each time step $t$, the encoder $E$ takes the context $\mathbf{C}^{(t)} = [\mathbf{x}_{t-1-w}, \ldots, \mathbf{x}_{t-1}]$ from a local window $W$ and the current logits $\mathbf{l}^{(t)}$ from the on-the-fly LLM $M$ to generates the watermark logits $\mathbf{l}_W^{(t)} = E(\mathbf{C}^{(t)}, \mathbf{l}^{(t)})$. The next token is then generated with the perturbed logits $\hat{\mathbf{l}}^{(t)} = \mathbf{l}^{(t)} + \delta \cdot \mathbf{l}_W^{(t)}$ based on Gumbel-Softmax sampling to enable differentiable sampling for end-to-end training. Once the stop criteria are reached, the entire watermarked sequence $\mathbf{X}_{\text{wm}}$ is generated, and the online text editor $N$ augments the text to simulate user edits, resulting in $\hat{\mathbf{X}}_{\text{wm}} = N(\mathbf{X}_{\text{wm}})$, which is then passed to the decoder $D$ to detect the presence of the watermark $\hat{\mathbf{m}} = D(\hat{\mathbf{X}}_{\text{wm}}) \in \{0, 1\}$, where "0" indicates no watermark and "1" indicates the presence of the watermark. The same prompt $\mathbf{X}_{\text{prompt}}$ is fed into the standard LLM

pipeline with $M$ to generate the non-watermarked sample $\mathbf{X}_{\text{nwm}}$ for $D$. The networks are jointly trained, with $E$ updated via backpropagation from $D$. After the model is trained, a converter $C$ is appended before both $E$ and $D$ for cross-LLM inference.

## 3.2 Watermark Embedding

We illustrate the watermark embedding process of our method, as shown in Fig. 1 (a) colored with light blue. Similar to KGW, our approach embeds a watermark on the generated text by biasing the original logits. Instead of using context token hashing to determine the bias, we employ a lightweight network $E$ to implicitly learn watermark logits by minimizing the detection loss $\mathcal{L}_{\text{dec}}$ for $D$ and the semantic loss $\mathcal{L}_{\text{sem}}$ between $\mathbf{X}_{\text{wm}}$ and $\mathbf{X}_{\text{nwm}}$. However, constructing such an end-to-end pipeline poses challenges due to the non-differentiable modules involved, including the token sampling process, the online editing of $\mathbf{X}_{\text{wm}}$, and the computation of $\mathcal{L}_{\text{sem}}$. To address this, we implement several alternatives, including using Gumbel-Softmax sampling to replace the non-differentiable sampling. Additionally, we propose an online prompting technique to perform online editing of $\mathbf{X}_{\text{wm}}$, as well as extract semantic embeddings to compute $\mathcal{L}_{\text{sem}}$ by dynamically prompting the on-the-fly LLM.

We now formulate the data flow of the watermark embedding process. At each time step $t$, the encoder $E$ receives the context $\mathbf{C}^{(t)}$ and the current logits $\mathbf{l}^{(t)}$ to generate the watermark logits $\mathbf{l}_W^{(t)}$ across the vocabulary $\mathcal{V}$. The encoder considers all possible token sequences $\mathbf{S}^{(t)} = [\mathbf{S}_1^{(t)}, \ldots, \mathbf{S}_{|\mathcal{V}|}^{(t)}]$, where each sequence is formed by $\mathbf{S}_i^{(t)} = [\mathbf{C}^{(t)}, \mathbf{t}_i]$. Nevertheless, due to the large size of $\mathcal{V}$, processing all sequences is computationally expensive. For efficiency, we focus on the top-$k$ logits tokens $\mathbf{t}_{i_1}, \ldots, \mathbf{t}_{i_k}$, and form the top-$k$ sequences $\mathbf{S}_{\text{top-}k}^{(t)} = [\mathbf{S}_{i_1}^{(t)}, \ldots, \mathbf{S}_{i_k}^{(t)}]$, in which $\{i_1, i_2, \ldots, i_k\}$ is the index of the top-$k$ logits tokens. A multilayer perceptron (MLP) $f_{\text{mlp}}$ maps the input sequence $\mathbf{S}_{\text{topk}}^{(t)}$ to the watermark logits of the top-$k$ tokens:

$$\mathbf{l}_{\text{top-}k}^{(t)} = \tanh(\tau_t \cdot f_{\text{mlp}}(\mathbf{S}_{\text{top-}k}^{(t)})), \tag{2}$$

where $\tanh$ bounds the output within $[-1, 1]$ and the parameter $\tau_t$ adjusts the sharpness of $\tanh$. As visualized by $\mathbf{l}_W^{(t)}$ in Fig. 1 (a), with logits of token close to -1 belonging to the "red" list and 1 indicating the "green" list. Except for the top-$k$ tokens, the watermark logits of the remaining tokens (can be considered in the "grey" list) are padded with 0 to form $\mathbf{l}_W^{(t)}$.

By adding up watermark logits on the original logits with strength $\delta$, we obtain the perturbed logits $\hat{\mathbf{l}}^{(t)}$. Then, the Gumbel-Softmax sampling (Jang et al., 2017) is used to allow the sampling step differentiable, mimicking the standard LLM sampling and enabling end-to-end training. Specifically, Gumbel noise $g_i = -\log(-\log(U_i))$, where $U_i \sim \text{Uniform}(0, 1)$, is added to each logit, yielding $\tilde{l}_i^{(t)} = \hat{l}_i^{(t)} + g_i$. The logits $\tilde{\mathbf{l}}^{(t)}$ are then passed through the softmax function:

$$\mathbf{p}_M(\mathbf{x}_t) = \frac{\exp((\hat{\mathbf{l}}^{(t)} + \mathbf{g})/\tau_g)}{\sum_{i=1}^V \exp((\hat{l}_i^{(t)} + g_i)/\tau_g)}, \tag{3}$$

where $\mathbf{g}$ is a vector of Gumbel noise with the same shape as $\hat{\mathbf{l}}^{(t)}$ and $\tau_g$ controls the sharpness of the softmax. The next token embedding is computed as $\mathbf{x}_t = \mathbf{p}_M^T \mathbf{E}$, where $\mathbf{E}$ is the token embedding matrix of $\mathcal{V}$. The watermark signal is embedded at each generation step until the stop criteria are met and the watermarked sample $\mathbf{X}_{\text{wm}}$ is obtained.

## 3.3 Online Text Editing

As shown in Fig. 1 (b), the online text editor $N$ (in light purple) is positioned between the encoder and decoder during training to simulate user edits of watermarked content, enhancing detection robustness. We utilize the online prompting technique in Fig. 2 (a) to augment watermarked text by dynamically prompting the online LLM $M$. This approach effectively handles non-differentiable online text modifications, such as rewriting. Specifically, after generates watermarked text $\mathbf{X}_{\text{wm}}$, which is fed into $N$ to produce $\hat{\mathbf{X}}_{\text{wm}}$. The editing prompt: *Rewrite the following paragraph:* `[text]` . *Now start to rewrite the above paragraph:* instructs $M$ to generate an augmented version of the watermarked content, where `[text]` is a placeholder for $\mathbf{X}_{\text{wm}}$. Thus, $N(\mathbf{X}_{\text{wm}})$ is equivalent to

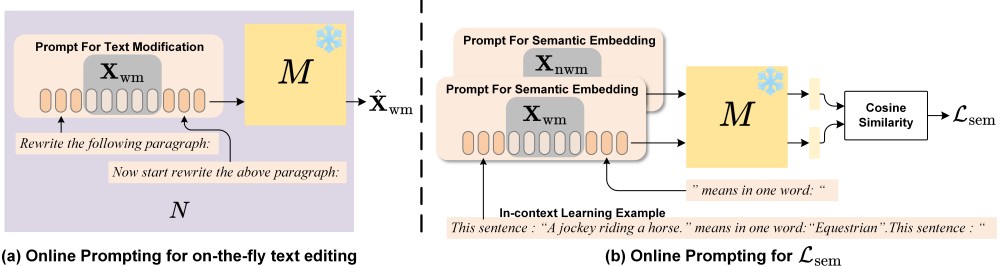

Figure 2: Online prompting technique for (a) computing semantic loss and (b) on-the-fly text editing. The prompts are first converted into the embedding domain and then concatenated with the generated text $\mathbf{X}_{\text{wm}}$ and $\mathbf{X}_{\text{nwm}}$.

$M([\mathbf{X}_{\text{epb}}, \mathbf{X}_{\text{wm}}, \mathbf{X}_{\text{epe}}])$ in which $\mathbf{X}_{\text{epb}}$ and $\mathbf{X}_{\text{epe}}$ denote the beginning and the end of the editing prompt. In contrast to existing online text editing methods, such as random token dropping/adding (Zhang et al., 2024), our approach shows significantly improved robustness to unseen distortions due to the capabilities of the on-the-fly $M$ to perform more complex modification on the text. While external paraphrasing models like Dipper (Krishna et al., 2023) are available, differences in tokenizers between the LLM and these models render the operation non-differentiable.

### 3.4 WATERMARK DETECTION

As shown in Fig. 1 (c), a lightweight network $D$ (in green) is used to classify whether a given text is watermarked or not, allowing end-to-end training for more robust detection compared to purely statistical methods. $\hat{\mathbf{X}}_{\text{wm}}$ (altered or unaltered) and $\mathbf{X}_{\text{nwm}}$ are fed into $D$, which predicts whether the text contains a watermark $\hat{m} = D(\mathbf{X})$. For efficiency, $D$ is built with LSTM layers and an MLP classification head.

### 3.5 TRAINING OBJECTIVES

The entire end-to-end system is supervised with two objectives: detection loss $\mathcal{L}_{\text{dec}}$ and semantic loss $\mathcal{L}_{\text{sem}}$. The detection loss $\mathcal{L}_{\text{dec}}$, which is computed using cross-entropy between the prediction and the ground-truth label to detect the watermark signal accurately. Additionally, to preserve the capabilities of the original LLM, $\mathcal{L}_{\text{sem}}$ ensures that $\mathbf{X}_{\text{wm}}$ retains the same semantics as $\mathbf{X}_{\text{nwm}}$. Semantic loss can be typically computed by the distance between embeddings which are extracted from an external semantic model $f_{\text{sem}}$, such that $\mathbf{e}_{\text{nwm}} = f_{\text{sem}}(\mathbf{X}_{\text{nwm}})$ and $\mathbf{e}_{\text{wm}} = f_{\text{sem}}(\mathbf{X}_{\text{wm}})$, and the loss can be computed by

$$\mathcal{L}_{\text{sem}}(\mathbf{e}_{\text{nwm}}, \mathbf{e}_{\text{wm}}) = 1 - \frac{\langle \mathbf{e}_{\text{nwm}}, \mathbf{e}_{\text{wm}} \rangle}{\|\mathbf{e}_{\text{nwm}}\|_2 \|\mathbf{e}_{\text{wm}}\|_2}. \tag{4}$$

Still, extracting the embeddings $\mathbf{e}_{\text{nwm}}$ and $\mathbf{e}_{\text{wm}}$ is non-differentiable. Unless the online LLM $M$ and the semantic model $f_{\text{sem}}$ share the same tokenizer, the embedding mapping between the two is not bijective, complicating the transformation from $M$ to the $f_{\text{sem}}$ domain. To address the issue, we avoid external models for computing semantics and instead prompt the on-the-fly LLM to approximate $f_{\text{sem}}$. Fortunately, recent studies have shown that LLMs can generate semantic embeddings with quality comparable to dedicated semantic models using prompt engineering, without the need for fine-tuning (Jiang et al., 2023). We leverage this LLM ability to compute $\mathcal{L}_{\text{sem}}$ directly, enabling end-to-end training and resolving non-differentiability issues. As illustrated in Fig. 2 (b), $\mathbf{X}_{\text{wm}}$ and $\mathbf{X}_{\text{nwm}}$ are generated in parallel at the generation phase, and a prompt with an in-context learning example: *This sentence: "A jockey riding a horse." means in one word: "Equestrian". This sentence: "[text]" means in one word: "* guides the on-the-fly LLM to extract semantic embeddings. `[text]` is the placeholder for $\mathbf{X}_{\text{nwm}}$ or $\mathbf{X}_{\text{wm}}$, and the embeddings $\mathbf{e}_{\text{nwm}} = M([\mathbf{X}_{\text{spb}}, \mathbf{X}_{\text{nwm}}, \mathbf{X}_{\text{spe}}])$ and $\mathbf{e}_{\text{wm}} = M([\mathbf{X}_{\text{spb}}, \mathbf{X}_{\text{wm}}, \mathbf{X}_{\text{spe}}])$ are extracted, where $\mathbf{X}_{\text{spb}}$ and $\mathbf{X}_{\text{spe}}$ are the beginning and the end of the above prompt. Finally, $\mathcal{L}_{\text{sem}}$ is computed by the cosine distance between the two embeddings as formulated in Eq. (4).

## 3.6 CROSS-LLM INFERENCE

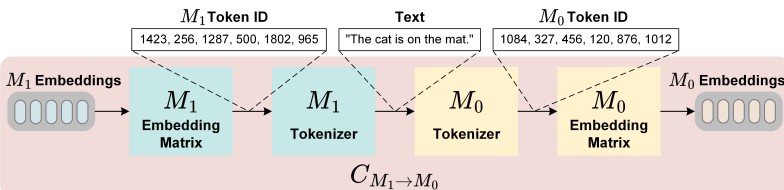

Figure 3: Converter for cross-LLM Inference.

After training $E^*_{M_0}$ and $D^*_{M_0}$ on a specific LLM $M_0$ using the end-to-end pipeline, we consider the generalizability of our method to other LLMs. Due to unique tokenizers and embedding dimensions, $E^*_{M_0}$ and $D^*_{M_0}$ cannot be directly applied to another LLM $M_1$. To address this, we introduce a converter $C$ that transforms embeddings from $M_1$ to $M_0$, as located in Fig. 1 (d) and detailed in Fig. 3. This process involves converting $M_1$ embeddings to the text domain and then back to $M_0$ embedding space. The converter is positioned before $E^*_{M_0}$ and $D^*_{M_0}$, enabling cross-LLM watermark embedding and detection as $E^*_{M_0}(C_{M_1 \to M_0}(\mathbf{S}))$ and $D^*_{M_0}(C_{M_1 \to M_0}(\mathbf{X}))$, respectively. Our encoder accepts a fixed number of context tokens, requiring careful token management due to the variability in token segmentation across different tokenizers. To ensure that the transformed context tokens exceed the original length, we dilate the context window $W$ by a factor of 2 and then truncate the latest context to maintain the appropriate length $W$.

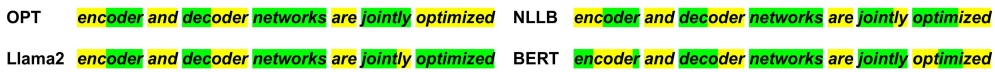

Figure 4: Visualize the tokenization of OPT (Zhang et al., 2022), Llama2 (Touvron et al., 2023), NLLB Costa-jussà et al. (2022), and BERT Kenton & Toutanova (2019) tokenizers. Each single token is indicated with a color block alternatively.

To analyze the applicability of converter $C$ for cross-LLM inference, we visualize the tokenization of 4 tokenizers in Fig. 4, and observe that the tokenizers often decompose sentences similarly. The converter $C$ processes candidate sequences $\mathbf{S}^{(t)}$ (for the encoder) and sentence $\mathbf{X}$ (for the decoder), which $\mathbf{S}^{(t)}$ include the context $\mathbf{C}^{(t)}$ and the next token candidate $\mathbf{t}_i$. Analysis of the transformation reveals that $\mathbf{X}$ and $\mathbf{C}^{(t)}$, containing numbers of tokens, can tolerate token variations. The transformed $\mathbf{t}_i$ can result in three outcomes: 1) merging with the preceding tokens (e.g., "der" merges with "deco" to form "decoder"); 2) splitting into sub-tokens (e.g., "jointly" becomes "joint" and "ly"); and 3) remaining unchanged. While the third outcome is ideal, cases 1 and 2 may negatively impact performance of the watermark model.

Table 1: Normalized Levenstein similarity between tokenized sentences across 4 tokenizers.

| Tokenizer | OPT | Llama2 | NLLB | BERT |
|---|---|---|---|---|
| OPT | 1.000 | 0.711 | 0.723 | 0.838 |
| Llama2 | - | 1.000 | 0.730 | 0.681 |
| NLLB | - | - | 1.000 | 0.721 |

To estimate the occurrence probability of cases 1 and 2, we quantify the alignment using normalized Levenshtein similarity between token lists for the same sentences across tokenizers, as shown in Table 1. The average token alignment probability is 73.4%. We find that given a large enough number of candidates $k$, the converter can be effectively utilized for cross-LLM inference [1], as demonstrated in our experiments.

---

[1] Given $k = 20$, we calculate that the probability of more than half of the candidate tokens being misaligned is 0.7% using the binomial distribution.

# 4 EXPERIMENTS

In this section, we present our experimental results, starting with the setup. We then compare watermark detection accuracy and text quality with SOTA methods, evaluate the undetectability and efficiency, and follow with ablation studies.

## 4.1 EXPERIMENT SETTINGS

### 4.1.1 IMPLEMENTATION DETAILS

To train our end-to-end model, we chose OPT-1.3B as the online LLM for efficiency. We use samples from the WikiText-103 dataset (Merity et al., 2017) as prompts during training. The context window is set to be $W = 10$, with the watermark strength $\delta = 1.25$ and $k = 20$ for the top-$k$ watermark logits empirically. We employ MGDA (Huo et al., 2024; Désidéri, 2012) to balance detection and semantic losses and use the Adam optimizer with a learning rate of 1e-4 over 35k training steps. All experiments are conducted on the NVIDIA RTX A6000 48G GPUs. The evaluation is performed based on the MARKLLM (Pan et al., 2024), an open-source tool for benchmarking LLM watermark methods. For further implementation details, please refer to our code which is available in supplementary material.

### 4.1.2 METRICS

We benchmark LLM watermarking methods across five key aspects, following the tradition of Pan et al. (2024); Zhang et al. (2024): 1) detection effectiveness of the human-written and pristine watermarked text, measured by the F1 scores at optimal thresholds; 2) detection robustness, evaluated by subjecting the watermarked text to 6 types of text modifications; 3) text quality, assessed using metrics including text perplexity and log diversity, as well as performance on downstream tasks including translation and code generation; 4) undetectability, by visualize token distribution and training BERT classifiers to distinguish between watermarked and non-watermarked samples; and 5) efficiency of the watermarking model, by measuring both the time and GPU memory overhead for text generation and watermark detection.

### 4.1.3 COMPETITORS

We compare our method with the SOTA logits-based methods: KGW(Kirchenbauer et al., 2023), Unigram(Zhao et al., 2024), SWEET(Lee et al., 2023), UPV(Liu et al., 2024a), SIR(Liu et al., 2024b) and TSW(Huo et al., 2024), and sampling-based methods EXP(Kuditipudi et al., 2024).

## 4.2 WATERMARK EFFECTIVENESS AND ROBUSTNESS

The benchmarking results of effectiveness and robustness are presented in Fig. 5, which compares 8 methods across 6 types of text modifications with 3 LLMs[2]. We used the first 30 tokens of samples from the C4 dataset (Raffel et al., 2020) as prompts, generating 200 tokens as watermarked samples, with the original human-written text serving as non-watermarked samples. As shown in the 'No Edit' column of Fig. 5, where the watermarked text remains unaltered, our method achieves competitive results with F1 scores of 1.000, 0.998, and 0.909 across the LLMs. To assess robustness, we evaluated the methods against 6 types of text modifications, including word deletion, synonym substitution, context-aware synonym substitution, and Dipper paraphrasing from Pan et al. (2024), along with emoji and copy-paste attacks from Liu et al. (2024b). Our model achieves the highest average F1 scores against the logits-based methods, with consistent performance gains of 1.2%, 4.0%, and 5.5% for the 3 LLMs, compared with the second-best method, Unigram. Notably, despite our model being trained on OPT-1.3B and the online text editor only including the paraphrasing prompt, it demonstrates superior generalizability to other LLMs and shows strong robustness against a wide

---

[2]We strictly follow the method settings, which can impact the balance between quality and accuracy, especially for logits-based approaches that are sensitive to the watermark strength $\delta$. Comparing the output distribution entropy for three LLMs, we find that at a fixed temperature, $T = 1$, the entropy of Llama2-7B-Chat is about half that of the other two LLMs, resulting in degraded detection performance of the logits-based methods. For a fair comparison, we focus on our method against logits-based approaches, while also reporting the sample-based method, EXP, for reference.

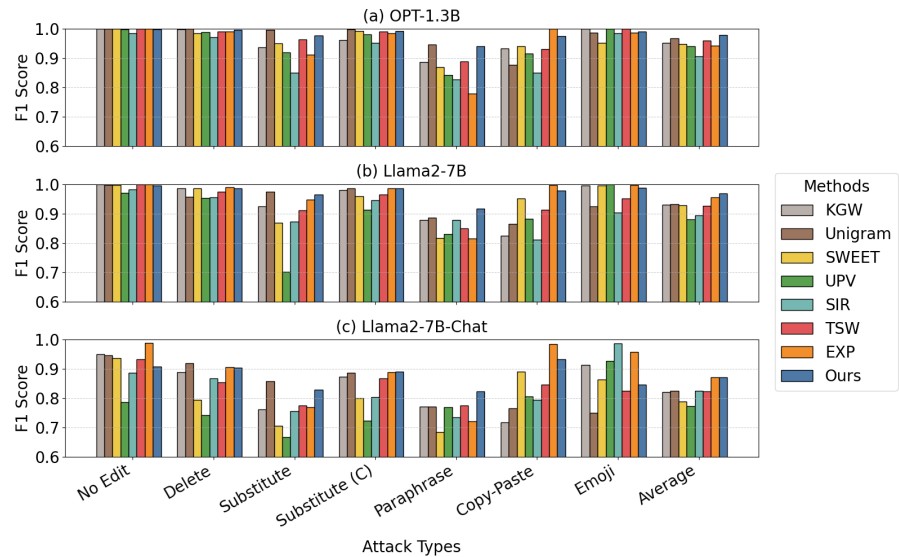

Figure 5: Detection accuracy (F1 score) on OPT-1.3B, Llama2-7B, and Llama2-7B-Chat for both pristine and edited watermarked samples.

range of unseen distortions. Although Unigram also achieves competitive detection accuracy and robustness, as will be clear soon, it introduces significant bias in the watermarked text, which substantially shifts the token distribution.

## 4.3 TEXT QUALITY

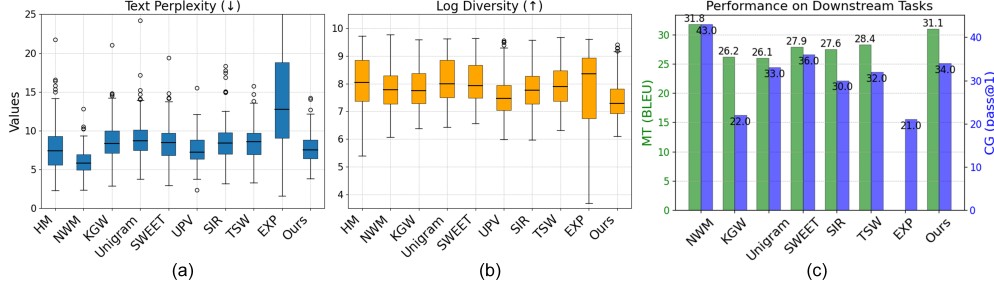

Figure 6: Quality assessment of watermarked sentences with (a) text perplexity, (b) log diversity, as well as (c) machine translation and code generation. HM and NWM are abbreviations of human-written and non-watermarked sentences.

To assess the impact of the watermark on text quality, we compute text perplexity (using Llama2-13B as the oracle model) and log diversity for text generated by Llama2-7B. We also evaluate the influence of the watermark on downstream tasks, including machine translation with the WMT16 German-English dataset (Bojar et al., 2016) using NLLB-200-distilled-600M (Costa-jussà et al., 2022), and code generation with the HumanEval (Chen et al., 2021) dataset using Starcoder (Li et al., 2023). EXP and UPV are excluded from corresponding tasks due to incompatibility. Fig. 6 (a) and (b) show that UPV achieves the lowest average perplexity of 7.531, closely followed by our method, which slightly increases the value by 2.6%, making it the second-best. Both methods apply perturbations to the top-$k$ logits, promoting the sampling of high-logit tokens and resulting in low perplexity, though this also leads to reduced log diversity. Importantly, low log diversity does not adversely affect LLM performance on downstream tasks. Fig. 6 (c) highlights that our method achieves the highest BLEU score of 31.062 in translation, surpassing the second-best method, TSW by 9.5%. For code generation, we attain a pass@1 score of 34.0, with the method dedicated to

code generation, SWEET, slightly higher at 36.0, which demonstrates the effectiveness of our approach in low-entropy scenarios. Compared to Unigram, which demonstrates competitive detection robustness, our method outperforms it by 19.2% and 3.03% on the two tasks, respectively.

## 4.4 UNDETECTABILITY

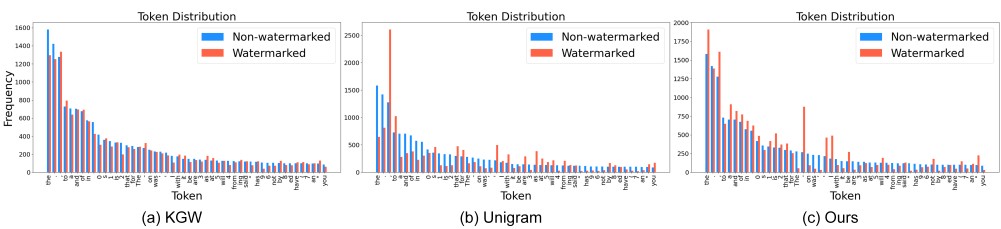

Figure 7: Token distribution between non-watermarked/watermarked sentences across three methods: (a) KGW, (b) Unigram, and (c) Ours.

We conduct two experiments to evaluate the undetectability of the methods. Firstly, we visualize the distribution of the most frequent tokens to reveal the token bias in the watermarked sentences. Then, we trained a classifier to distinguish between watermarked and non-watermarked sentences.

**Token Distribution.** In Fig. 7, we present the top 50 most frequent tokens from the original Llama2-7B generated texts, together with the watermarked texts across three methods. Fig. 7 (a) shows that KGW has the most similar token distribution to the original text; however, as discussed in previous sections, the detection accuracy and robustness are relatively low. The Unigram method, depicted in Fig. 7 (b), achieves competitive detection performance through global red-green list splitting but at the cost of significant token bias that makes it detectable. Our approach, shown in Fig. 7 (c), strikes the best balance by achieving high detection accuracy while minimally impacting the performance of LLMs, introducing moderate bias in the distribution. We hypothesize that this balance is due to our method concentrating the bias on low-frequency tokens while preserving the distribution of high-frequency tokens, as observed in the token distribution.

Table 2: Accuracy of classifying watermarked and non-watermarked sentences.

| Method | BERT-base | | BERT-large | | Method | BERT-base | | BERT-large | |
|---|---|---|---|---|---|---|---|---|---|
| | ACC | F1 | ACC | F1 | | ACC | F1 | ACC | F1 |
| KGW | 0.460 | 0.471 | 0.460 | 0.631 | SIR | 0.740 | 0.755 | 0.720 | 0.774 |
| Unigram | 0.860 | 0.877 | 0.660 | 0.721 | TSW | 0.600 | 0.565 | 0.400 | 0.571 |
| SWEET | 0.440 | 0.588 | 0.520 | 0.684 | EXP | 0.700 | 0.689 | 0.580 | 0.000 |
| UPV | 0.540 | 0.676 | 0.480 | 0.000 | Ours | 0.680 | 0.570 | 0.560 | 0.267 |

**Training Deep Classifier.** Moreover, we train two classifiers, BERT-base, and BERT-large (Kenton & Toutanova, 2019), on 400 non-watermarked/watermarked samples [3]. We used another 200 samples for testing. The detection accuracy and F1 score of the two classifiers among the watermark methods are shown in Table 2. We can observe that the two classifiers are confused in distinguishing the watermarked samples generated by our method, with only 0.570 and 0.267 F1 scores respectively. Unigram, which introduces remarkable bias on the token distribution, is found to have relatively higher F1 scores, 0.877 and 0.721 across the two classifiers.

## 4.5 WATERMARK EFFICIENCY

In Table 3, we assess the computational time overhead and GPU memory usage of our method. For OPT-1.3B, the lightweight encoder design results in only an 8.3% increase in generation time for

---

[3]The prompt between non-watermarked and watermarked texts are different (Zhang et al., 2024). Since the attacker can only get the watermarked text from the protected LLM API and gather other unwatermarked text from other places.

Table 3: Time and memory consumption for generation and detection with 200 tokens of our method.

| LLM | Setting | Generation | | Detection | |
|---|---|---|---|---|---|
| | | Time (s) | Memory (GB) | Time (s) | Memory (GB) |
| OPT-1.3B | w/o WM | 2.557 | 5.900 | 0.003 | 0.008 |
| OPT-1.3B | w/ WM | 2.769 | 5.900 | 0.003 | 0.008 |
| Llama2-7B | w/o WM | 5.204 | 15.793 | 0.005 | 0.008 |
| Llama2-7B | w/ WM | 8.362 | 15.793 | 0.005 | 0.008 |

watermarked text, with no change in maximum GPU memory usage since the encoder is invoked after each token generation. For Llama2-7B, our method increases the generation time by 60.6%, mainly due to the embedding transformation from Llama to OPT, as the tokenizer cannot be accelerated by the GPU and is called at each step. In terms of watermark detection, our decoder operates efficiently, requiring only negligible time and memory consumption.

## 4.6 ABLATION STUDY

Table 4: Detection performance and quality assessment in different settings of our method.

| Setting | Accuracy | | | Quality | |
|---|---|---|---|---|---|
| | No Attack | Paraphrase (Dipper) | Copy-Paste | Perplexity $\downarrow$ | Translation BLEU $\uparrow$ |
| w/o $\mathcal{L}_{\text{sem}}$ | 0.995 | 0.960 | 0.982 | 9.561 | 30.155 |
| w/o $N$ | 0.992 | 0.867 | 0.952 | 7.820 | 31.063 |
| $\delta = 1$ | 0.990 | 0.912 | 0.949 | 7.628 | 31.279 |
| $\delta = 1.5$ | 1.000 | 0.970 | 0.978 | 8.087 | 30.676 |
| $\delta = 1.75$ | 1.000 | 0.977 | 0.982 | 8.684 | 30.279 |
| $\delta = 2$ | 1.000 | 0.980 | 0.983 | 9.320 | 30.001 |
| Ours ($\delta = 1.25$) | 0.998 | 0.941 | 0.975 | 7.730 | 31.062 |

We conduct the ablation studies to evaluate the impact of various settings of our method, including the semantic loss $\mathcal{L}_{\text{sem}}$, the online text editor $N$, and the watermark strength $\delta$. The results of this analysis are summarized in Table 4, which presents detection performance alongside quality metrics. The first row of the table reveals that removing the semantic loss supervision $\mathcal{L}_{\text{sem}}$ leads to a significant increase of 23.7% in the perplexity. This suggests that the inclusion of online prompting $\mathcal{L}_{\text{sem}}$ plays a crucial role in maintaining the quality of the generated text. Additionally, the second row highlights the importance of the online text editor $N$. Without this module, the performance of the decoder diminishes, resulting in a 7.9% and 2.4% decline in F1 scores for the paraphrasing and copy-paste attack. We also investigate the effect of varying watermark strength $\delta$. It indicates a significant trade-off between detection robustness and text quality. When $\delta$ is set too small (e.g., $\delta = 1$), we observe a reduction in detection performance, evidenced by an average F1 score decline of 2.2%. On the other hand, when $\delta$ is excessively large (e.g., $\delta = 2$), the quality of the text deteriorates, as indicated by an increase in perplexity by 20.6%. This balance is critical, as our method aims to optimize both detection efficacy and the quality of the generated text.

## 5 CONCLUSION

We introduce the first logits-based end-to-end model, where encoder and decoder networks are jointly optimized to improve detection robustness and text quality. Constructing such a system is challenging due to the non-differentiability of key modules, such as the online text editor and semantic loss computation. To overcome these obstacles, we propose a novel online prompting technique that leverages the on-the-fly LLM as a differentiable surrogate, effectively handling these non-differentiable operations. Our method can be easily generalized to different LLMs.

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

# A ROBUSTNESS QUALITY TRADE-OFF

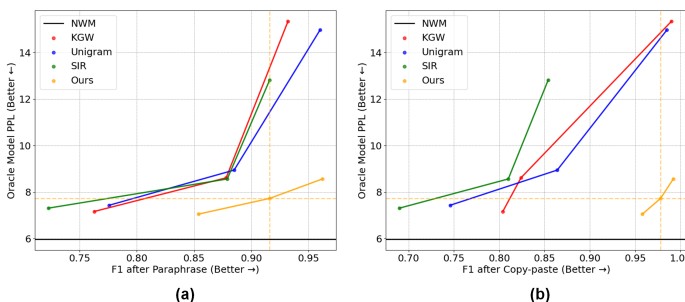

Figure 8: The trade-off between F1 score after text modifications and Llama2-13B perplexity. (a) paraphrasing; (b) copy-paste attack.

We evaluate our model on Llama2-7B to demonstrate its superior balance between robustness and quality compared to SOTA methods. Since our model is trained on OPT-1.3B, this serves as a cross-LLM evaluation. The watermarked text is subjected to two common modifications: paraphrasing and copy-paste attacks. Robustness and quality are analyzed at various watermark strengths[4] $\delta$, as shown in Fig. 8.

The watermark strength $\delta$ for each method is set as follows:

- KGW and Unigram: $\delta = 1, 2$(default), $5$
- SIR: $\delta = 0.5, 1$(default), $2$
- Our model: $\delta = 0.75, 1.25$(default), $1.5$

Our model consistently achieves a superior trade-off between robustness and quality compared to other methods. The perplexity (PPL) of non-watermarked text is approximately 6, represented by the black horizontal line as a reference. At the PPL level below 8, our model demonstrates significantly higher robustness under both paraphrasing and copy-paste attacks, outperforming the second-best model by margins of **18.16%** and **20.42%**, respectively. In contrast, existing methods, KGW, Unigram, and SIR achieve increased robustness at the expense of quality, resulting in PPL that is $2\times$ to $3\times$ higher than those of non-watermarked text, leading to significant degradation in text quality.

Table 5: Comprehensive performance of SOTA LLM watermarking methods. NA, SC, PP, and CP are abbreviations of no attack, context-aware synonym substitution, paraphrasing, and copy-paste attack. PPL, LD, MT, and CG are abbreviations of perplexity, log diversity, machine translation, and code generation tasks. The best result is in **bold**, and the second-placed is underlined.

| Method | Accuracy/Robustness | | | | Quality | | | |
|---|---|---|---|---|---|---|---|---|
| | NA↑ | SC↑ | PP↑ | CP↑ | PPL↓ | LD↑ | MT↑ | CG↑ |
| KGW($\delta = 2$) | **1.000** | 0.980 | 0.878 | 0.824 | 8.615 | 8.077 | 26.200 | 22.0 |
| Unigram($\delta = 2$) | 0.997 | **0.985** | 0.885 | 0.864 | 8.948 | **8.836** | 26.061 | 33.0 |
| SIR($\delta = 1$) | 0.982 | 0.945 | 0.879 | 0.810 | 8.566 | 8.269 | 27.557 | 30.0 |
| Ours($\delta = 1.25, k = 20$) | 0.997 | **0.985** | 0.916 | 0.978 | **7.730** | 7.594 | 31.062 | **34.0** |
| Ours($\delta = 1.25, k = 40$) | 0.995 | 0.980 | **0.931** | **0.983** | 8.346 | 8.103 | **31.114** | **34.0** |

To comprehensively benchmark our model's performance against SOTA competitors, we present additional metrics in Table 5. For better visualization, Fig. 9 displays a radar graph with all metrics normalized using z-scores. Overall, our model (represented in red and purple) demonstrates superior performance, covering a larger area on the chart compared to competitors. Notably, with $\delta = 1.25$

---

[4]The strength of each model is set to its original value, with variations including one lower and one higher strength.

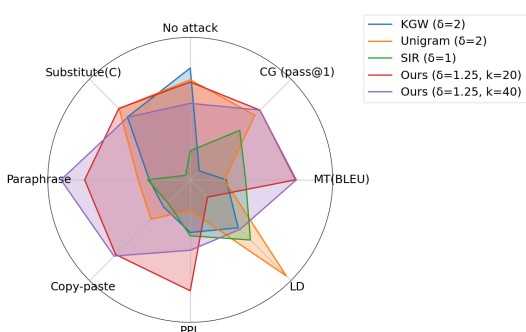

Figure 9: Radar chart comparing our models' performance against SOTA competitors across key accuracy and quality metrics.

and $k = 20$, our model achieves significantly better results across multiple dimensions: a **5.33% improvement in robustness**, a **9.76% reduction in PPL**, and a **7.88% increase in downstream task performance**. By holding $\delta$ constant and varying the top-$k$ values to 40, our model effectively improves log diversity (LD). This adjustment increases perplexity as a trade-off while having only a minor impact on other metrics. This flexibility allows users to fine-tune the trade-offs between text quality and diversity to match their specific requirements. It is observed that all the methods cannot achieve both low PPL and high LD simultaneously due to the fundamental trade-offs. Low PPL requires concentrating probability on a few high-likelihood tokens while high LD demands spreading probability across a wider range of tokens to encourage variety. This broader distribution increases unpredictability, which can raise PPL.

## B  FALSE POSITIVE THRESHOLDING

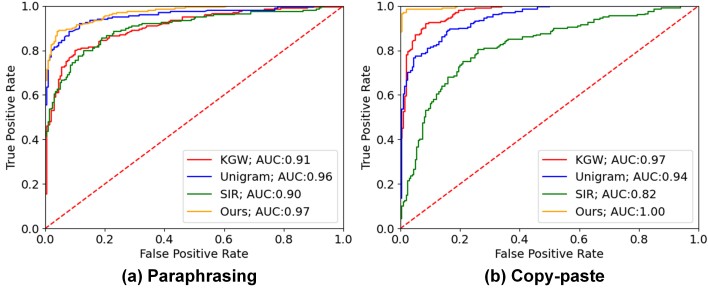

Figure 10: ROC curves for watermark detection under (a) paraphrasing and (b) copy-paste attack.

Unlike statistical decoding, which relies on the p-value, our model utilizes a deep classifier that produces a scalar confidence score for watermark detection. Similarly, both p-value-based and our method use a fixed threshold to detect the presence of a watermark, which also allows our decoder to be adjusted to control false positive rates. Figure 10 presents the ROC curves for watermark detection under various text modifications, illustrating our model's ability to effectively manage false positive rates through the threshold.

## C  WATERMARK SAMPLE VISUALIZATION

Figure 11 presents three pairs of non-watermarked and watermarked samples generated from the same prompt of our method. The tokens are colored based on the output values retrieved from the watermark encoder. The watermarked samples maintain high quality while featuring a greater

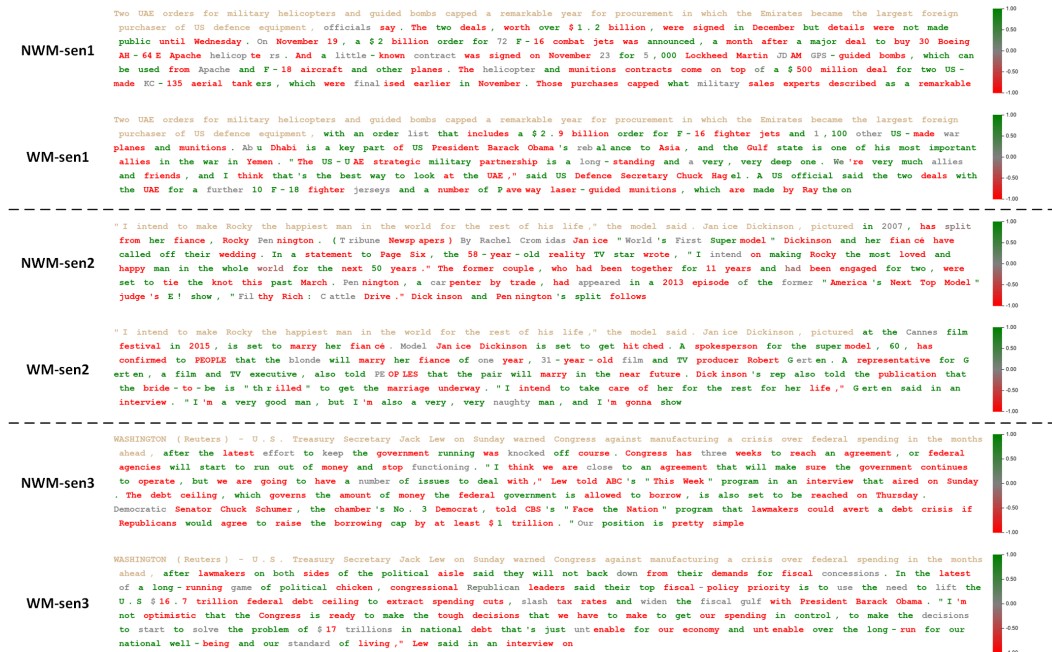

Figure 11: Visualizing three pairs of sentences comparing outputs from the original LLM (NWM) and the watermarked text (WM) of our method using the same prompt. Prompts are highlighted in gold, tokens are colored by the output values of the watermark encoder, and tokens outside the top-$k$ logits are shown in grey.

proportion of green-list tokens compared to red-list tokens. This is achieved by increasing the probabilities of green-list tokens during the generation process.

# D  TOKEN DISTRIBUTION BIAS

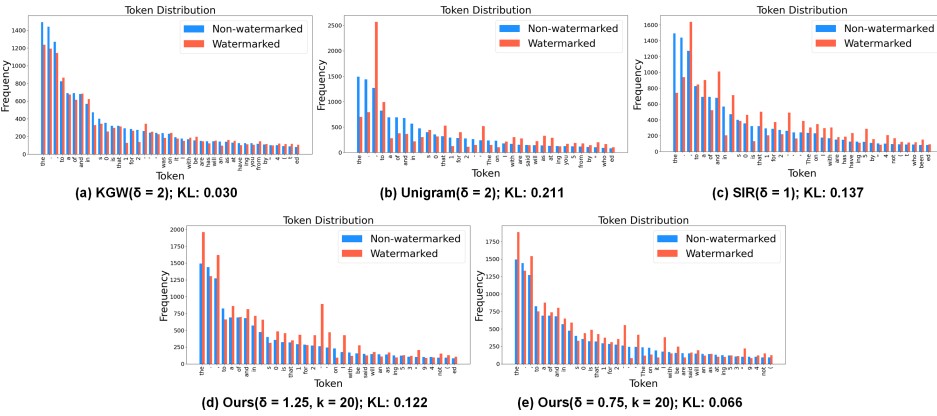

Figure 12: Token distribution and KD divergence (lower the better) between non-watermark/watermark sentences of our method in different settings as well as the SOTA competitors.

We further examine the effect of watermark strength $\delta$ on token distribution, as depicted in Fig. 12. With the default setting of $\delta = 1.25$, our model introduces less token bias compared to Unigram and SIR, while achieving superior robustness, as shown in Table 5. However, although our method outperforms KGW in robustness, it introduces a greater token bias. We argue that this highlights an inherent trade-off between robustness and token bias. Reducing $\delta$ to 0.75 reduces the KL divergence

by 50% compared to the default setting, but comes at the expense of reduced robustness. Figure 8 presents the corresponding F1 scores with the setting $\delta = 0.75$, illustrating this trade-off.

# E    WATERMARK DETECTABILITY WITH PAIRED SAMPLES

Table 6: Detection accuracy of paired non-watermark/watermark sentences with BERT-base against the size of the training set.

| Method | Training Size | | | | |
|---|---|---|---|---|---|
| | 100 | 200 | 500 | 1500 | 3000 |
| KGW($\delta = 2$) | 0.572 | 0.621 | 0.641 | 0.722 | 0.823 |
| Unigram($\delta = 2$) | 0.627 | 0.771 | 0.929 | 0.947 | 0.960 |
| SIR($\delta = 1$) | 0.603 | 0.590 | 0.806 | 0.900 | 0.942 |
| Ours($\delta = 1.25$, $k = 20$) | 0.599 | 0.617 | 0.778 | 0.888 | 0.893 |

We further train a BERT-base classifier using paired non-watermarked and watermarked samples generated from identical prompts with the same LLM. We argue that such paired samples are unlikely to be accessible to adversaries, as they can only request watermarked text from the LLM API and search for unwatermarked text online. However, if paired outputs are obtained using identical prompts from different LLMs, a detector could exploit the domain gap between the two LLMs to achieve high accuracy with trivial solutions. For reference, we conducted an experiment using such paired non-watermarked and watermarked samples generated by the same LLM. The training set sizes ranged from 100 to 3,000 samples, with equal numbers of watermarked and non-watermarked samples. The test set consisted of 1,000 watermarked and non-watermarked samples, and the detection accuracy is shown in Table 6. Overall, paired samples significantly increase the detectability of watermarked text compared to unpaired samples (as reported in Table 2), with detection accuracy improving as the training size grows. Among the methods tested, KGW showed slightly better undetectability compared to others, followed by our method. In contrast, watermarked samples from Unigram and SIR are nearly fully classified, with detection accuracies of 96.0% and 94.2%, respectively.

# F    MODEL TRAINING DETAILS

Table 7: Hyperparameters of the end-to-end model training.

| Item | Value |
|---|---|
| Learning rate | 1-e4 |
| Batch size | 16 |
| Training step | 35000 |
| Encoder context size $w$ | 10 |
| On-the-fly LLM | OPT-1.3B |
| Top-$k$ candidate | 20 |
| Prompt tokens | 30 |
| Gumbel-softmax temperature $\tau_g$ | 0.1 |
| Probability of activating $N$ | 0.5 |
| Sharpness of $\tanh$, $\tau_t$ | 1000 |
| Maximum generated tokens | 100 |
| Watermark strength $\delta$ | 1 |
| Weight of $\mathcal{L}_{\text{dec}}$ | 10 |
| Weight of $\mathcal{L}_{\text{sem}}$ | 1 |

We trained our end-to-end model on a single NVIDIA RTX A6000 48GB GPU for 35k steps, completing the training in approximately 5 days with a GPU memory usage of 21.96GB. The hyperparameters used during training are detailed in Table 7. If GPU memory is limited, the batch size and

maximum generated tokens can be reduced, or a smaller LLM, such as OPT-125M, can be used. While the training phase takes longer compared to existing training-based models like SIR, UPV, and TSW, this is due to the inclusion of the entire LLM (with frozen parameters) in the training process. Despite the longer training time, we developed an efficient converter for cross-LLM inference, ensuring that the computational cost during inference remains low.

## G   ATTACK CONFIGURATION

To evaluate robustness, we compare 8 methods across 6 types of attacks in Fig. 5. The setting of all 6 attacks strictly follow the open-source studies, MARKLLM[5] and SIR[6]. For transparency, we provide a detailed description, parameter setting, and prompt setting for each text modification.

Table 8: Details of the attacks in the robustness evaluation.

| Attack | Description | Parameters |
|---|---|---|
| Delete | Randomly drop words from the watermarked text. | Drop ratio: 0.3 |
| Substitute | Random synonym substitution from WordNet (Miller, 1995). | Replace ratio: 0.5 |
| Substitute (In-context) | Using BERT-large (Devlin, 2018) to select synonyms that fit the context. | Replace ratio: 0.5 |
| Paraphrase | Paraphrasing the text while maintaining the original meaning with deep paraphrase model Dipper(Krishna et al., 2023). | Lex diversity: 60 Order diversity: 0 |
| Copy-paste | Concatenating the watermarked text after the human-written text creates a mix with only parts of the watermarked text. | Watermarked text fraction: 25% |
| Emoji | The prompt is prefixed with the additional prompt: *inserting an asterisk * between each generated token*, and remove the asterisks from the generated watermarked text. This process disrupts the original preceding tokens of the watermarked text, making it more challenging for detection. | - |

## H   COMPARISON WITH POST-GENERATION WATERMARKING METHODS

Post-generation watermarking methods, such as AWT (Abdelnabi & Fritz, 2021) and REMARK-LLM (Zhang et al., 2024), embed watermarks after the text has been fully generated. These approaches rely on a language model to rephrase the generated text, embedding a watermark signal while preserving the semantic meaning of the original sentences. However, post-generation methods have notable limitations. They do not fully leverage the capabilities of the original LLM and are more susceptible to OOD issues. For instance, models trained on datasets like HC3, a natural language question-answering corpus, often struggle with OOD inputs, such as code, leading to reduced performance on tasks like code generation (e.g., lower code passing scores). In contrast, logits-based methods, including ours, embed watermarks during the generation process by sampling tokens directly from a perturbed distribution. This approach minimally constrains the LLM, allow-

---

[5]https://github.com/THU-BPM/MarkLLM
[6]https://github.com/THU-BPM/Robust_Watermark

ing it to retain its natural understanding of language while maintaining broad compatibility across diverse tasks.

## I WATERMARK DETECTION PERFORMANCE WITH BEAM SEARCH

We evaluated the effectiveness and robustness of watermarking methods across two LLM decoding strategies, including multinomial sampling and beam search (5 beams). The detection performance (F1 score) is summarized in Table 9. Our method achieves the highest average F1 score across both decoding strategies, with slightly better performance in the beam search case. This improvement is attributed to beam search's preference for higher probability tokens, which implicitly selects more green-list tokens in the generated text.

Table 9: Detection performance with multinomial sampling and beam-search on LLama2-7B. NA, SC, PP, and CP are abbreviations of no attack, context-aware synonym substitution, paraphrasing, and copy-paste attack. The best result is in **bold**, and the second-placed is underlined.

| Method | Multinomial sampling | | | | Beam search (num_beams = 5) | | | |
|---|---|---|---|---|---|---|---|---|
| | NA | SC | PP | CP | NA | SC | PP | CP |
| KGW($\delta = 2$) | **1.000** | 0.980 | 0.878 | 0.824 | **1.000** | 0.998 | 0.940 | 0.975 |
| Unigram($\delta = 2$) | 0.997 | **0.985** | 0.885 | 0.864 | **1.000** | **1.000** | 0.954 | 0.939 |
| SIR($\delta = 1$) | 0.982 | 0.945 | 0.879 | 0.810 | 0.992 | 0.977 | 0.912 | 0.790 |
| Ours($\delta = 1.25$, $k = 20$) | 0.997 | **0.985** | **0.916** | **0.978** | 0.997 | 0.995 | **0.962** | **0.995** |

## J STATISTICALLY SIGNIFICANT IMPROVEMENTS

To validate the statistical significance of our model's improvements, we conducted robustness and quality experiments using a paired t-test. Specifically, we performed five trials to evaluate robustness against paraphrasing (PP) and copy-paste (CP) attacks, as well as the perplexity (PPL) of watermarked text. All trials were conducted using the same set of prompts, with multinomial sampling (same seed across methods in the same trial for consistency), resulting in diverse outcomes across the five trials. The watermark strength parameters ($\delta$) for KGW, Unigram, and our method were set to 2, 2, and 1.25, respectively, while the $k$-value for our method was set to 20. The results, shown in Table 10, indicate that our model achieves statistically significant improvements in both robustness and quality compared to the competitors. This is supported by p-values consistently below the 0.05 threshold.

Table 10: Paired t-test results comparing our model with competitors. Mean Diff. represents the difference between our model and each competitor's mean performance. All p-values below 0.05 indicate statistically significant improvements.

| Method | PP-Trials | | | | | Mean ± Std | Mean Diff. | p-value |
|---|---|---|---|---|---|---|---|---|
| KGW | 0.818 | 0.798 | 0.844 | 0.828 | 0.835 | 0.824 ± 0.018 | 0.018 | $2.12 \times 10^{-4}$ |
| Unigram | 0.910 | 0.899 | 0.892 | 0.906 | 0.882 | 0.898 ± 0.011 | 0.011 | $3.80 \times 10^{-2}$ |
| Ours | 0.918 | 0.908 | 0.909 | 0.906 | 0.917 | 0.912 ± 0.005 | - | - |
| Method | CP-Trials | | | | | Mean ± Std | Mean Diff. | p-value |
| KGW | 0.843 | 0.832 | 0.845 | 0.847 | 0.839 | 0.841 ± 0.006 | 0.006 | $5.51 \times 10^{-6}$ |
| Unigram | 0.864 | 0.900 | 0.859 | 0.861 | 0.866 | 0.870 ± 0.017 | 0.017 | $3.23 \times 10^{-4}$ |
| Ours | 0.968 | 0.960 | 0.964 | 0.967 | 0.942 | 0.960 ± 0.011 | - | - |
| Method | PPL-Trials | | | | | Mean ± Std | Mean Diff. | p-value |
| KGW | 8.656 | 8.678 | 8.800 | 8.730 | 8.409 | 8.655 ± 0.148 | -1.014 | $1.07 \times 10^{-4}$ |
| Unigram | 8.871 | 8.832 | 8.918 | 9.155 | 9.092 | 8.973 ± 0.142 | -1.333 | $3.82 \times 10^{-6}$ |
| Ours | 7.626 | 7.620 | 7.544 | 7.773 | 7.642 | 7.641 ± 0.083 | - | - |

