# OpenReview forum: "An End-to-End Model For Logits Based Large Language Models Watermarking"
_ICLR.cc/2025/Conference — Submitted to ICLR 2025_

### Official Review · Reviewer_JkAA · 2024-10-27

**Soundness:** 2
**Presentation:** 3
**Contribution:** 2
**Rating:** 3
**Confidence:** 5

**Summary:**

This paper proposed an end-to-end optimization framework for achieving better trade-off between the robustness and the text quality. The authors validate the effectiveness of the proposed framework with comprehensive experiments on popular LLMs.

**Strengths:**

1. The proposed end-to-end method is original, and extensive experiments have been conducted to evaluate its quality, detectability, and robustness.

2. The presentation is well-structured, making the paper easy to follow.

**Weaknesses:**

1. Unclear motivation. The authors claimed “However, these existing approaches still fail to achieve an optimal trade-off between text quality and robustness”. However, the authors have missed an important line of works regarding the distortion-free watermark (Kuditipudi et al., 2024; Christ et al., 2024), which suggested we can embed watermarks into LLMs without affect the generation quality. Thus, the there is generally no trade-off between the text quality and robustness, and the claim in the paper is wrong.

2. Limited contribution. Comparing to the previous works (Liu et al., 2024b; Huo et al., 2024), which also share an encoder-decoder structure for logits-based watermarking, the proposed method only introduce a jointly training network for achieving better trade-off between text quality and robustness. As the reviewer has pointed out in weaknesses 1, the trade-off generally does not exist. Thus, the contributions of the proposed method are unclear.

3. The experimental results also cannot support the motivation of “achieving better trade-off between text quality and robustness”. In Figure 7. The KGW watermark has significantly better quality than the proposed watermark, although the detectability and the robustness of KGW are poor. In order to claiming the proposed method has achieved better trade-off than KGW, the authors should show the superior of the proposed method on all quality, detectability, and robustness axis. Besides, in Figure 5, we can also see that the proposed method does not always outperform the baselines in all scenarios.

**Questions:**

N/A

---

> ### Author Response · Authors · 2024-11-22
>
> ### W1: Distortion-Free Watermarking
> - We have indeed considered distortion-free watermarking in our manuscript and described them as sampling-based methods (EXP). However, we argue that even distortion-free watermarking can degrade the quality of output text. This is supported not only by our empirical results in Fig. 6 ***but also by the LLM watermark benchmarking studies [A], [B]***. Consequently, we emphasize that the robustness/quality trade-off remains a persistent challenge for existing LLM watermarking models and is still an open problem to be solved in the field.
>
> ### W2: Contributions
> - Given the persistence of the robustness/quality trade-off, our primary contribution is achieving a better trade-off through encoder/decoder end-to-end training.
> - Additionally, we address the key challenge of incorporating non-differentiable modules into the end-to-end training pipeline by proposing an innovative online prompting approach.
> - As highlighted in Fig. 9, our model achieves a better performance across key metrics against SOTA competitors.
>
> ### W3: Robustness and quality Trade-offs
> - Please refer to the "Response to All Reviewers".  A detailed comparison of our method with SOTA competitors is presented in Table 5, accompanied by a radar graph in Fig. 9. These results demonstrate the clear advantages of our approach, including significant improvements in robustness, perplexity, and downstream task performance. This highlights our model's ability to achieve a well-balanced performance across key metrics.
>
> [A] Leyi Pan, Aiwei Liu, Zhiwei He, Zitian Gao, Xuandong Zhao, Yijian Lu, Binglin Zhou, Shuliang Liu, Xuming Hu, Lijie Wen, et al. Markllm: An open-source toolkit for llm watermarking. arXiv preprint arXiv:2405.10051, 2024
>
> [B] Piet, Julien, et al. "Mark my words: Analyzing and evaluating language model watermarks." arXiv preprint arXiv:2312.00273, 2023.

---

> ### Author Response · Authors · 2024-11-26
> **Request for Timely Feedback on our Clarification of Motivation and Contributions**
>
> Dear Reviewer JkAA,
>
> We apologize for reaching out again, but with the rebuttal discussion period concluding in **less than 36 hours**, ***Nov 26, 11:59 PM AoE***, we sincerely appreciate the continued time and effort you dedicate to discussing our submission.
>
> We justify our motivation with recent LLM watermark benchmarking studies, which confirm the existence of a robustness/quality trade-off in distortion-free watermarking methods. Our contributions are as follows: 1) addressing the robustness/quality trade-off through end-to-end training of the watermark encoder/decoder; 2) tackling the challenge of non-differentiable modules in the training pipeline with an innovative online prompting; and 3) demonstrating superior performance across key metrics compared to state-of-the-art competitors, as highlighted in Figure 9.
>
> We are truly grateful if you could let us know if there are any remaining questions or concerns about our submission. If our responses satisfactorily address your concerns, we kindly hope you might consider reflecting this in your review score.
>
> Thank you once again for your thoughtful feedback and for your consideration.
>
> Best regards,
> Submission9685 Authors

---

> ### Comment · Reviewer_JkAA · 2024-11-26
> **Keep my rating**
>
> Thanks for the rebuttal. My main concerns about the motivation and the contribution are not addressed. I decide to keep my rating.
>
> Distortion-free watermarks have a theoretical guarantee to preserve the text quality, the statement "even distortion-free watermarking can degrade the quality of output text" is not true. The degrade of the text quality observed in the related works could be caused by the different experimental settings. In Hu et al [1], the distortion-free watermark has the same generation quality of the output text as the original LM. Thus, the trade-off between the quality and detectability does not exist.
>
> Hu et al. Unbiased watermark for large language models, ICLR 2024

---

> > ### Author Response · Authors · 2024-12-02
> >
> > 1.**Robustness of sampling-based methods**
> >
> > To further validate our claim regarding the quality/robustness trade-off, we conduct an experiment on Unbiased alongside logits-based methods, including KGW, Unigram, and our proposed approach. The evaluation involved subjecting the watermarked text to Dipper paraphrasing under two settings, with the results shown below:
> >
> > | Method    | PP-Dipper (lex:60; order:0) | PP-Dipper (lex:60; order:20) |
> > |-----------|--------------------------|--------------------------|
> > | KGW       | 0.878                    | 0.792                    |
> > | Unigram   | 0.885                    | 0.879                    |
> > | Unbiased  | 0.687                    | 0.689                    |
> > | Ours      | **0.916**                    | **0.902**                    |
> >
> > The results demonstrate that our method achieves the highest F1 score among the competitors, followed by the other logits-based methods, KGW and Unigram. In contrast, the distortion-free method, Unbiased, performs poorly in detecting watermarked text after paraphrasing, with low F1 scores of 0.687 and 0.689 in the two paraphrasing scenarios, respectively.
> >
> > 2.**Influence of LLMs settings**
> >
> > We argue that a feasible watermarking scheme should demonstrate adaptability to various LLMs and configurations, rather than being effective under only specific conditions. These configurations may include variations in LLM itself, temperature, decoding strategies, top-$p$/top-$k$ values, or other parameters. ***An effective watermarking scheme should preserve both the quality of the LLM's output and the detection accuracy across diverse settings***.
> >
> > ***Our proposed method achieves this adaptability by leveraging the tunable watermark strength $\delta$ and top-$k$ values***. As shown in Fig. 5 and 6 of our manuscript, our approach consistently performs well across different LLMs and settings.
> >
> > Our experiments, along with prior studies such as SWEET (referenced in Table 1), reveal that sampling-based methods like EXP-Edit (which Unbiased also follows using a similar inverse transform principle) struggle to achieve high detection scores while maintaining distortion-free outputs. This limitation arises from the fixed nature of the sampling-based watermark schemes and the spiky distribution of low-entropy outputs generated by code-generation LLMs, which reduces the effectiveness of sampling-based approaches. To enhance detectability in low-entropy scenarios, increasing the temperature is often necessary; however, this adjustment compromises the quality of the generated output.

---

> ### Author Response · Authors · 2024-11-30
>
> We would like to clarify that our primary claim focuses on the trade-off between ***quality and robustness***, rather than ***quality and detectability***. In our work, we define detectability as the detection accuracy on unaltered, watermarked text; while robustness refers to the detection accuracy on modified watermarked text. The modifications may include edits made by the user after receiving the watermarked text, such as synonym substitution, paraphrasing, or other changes, which are common in practice.
>
> An ideal watermarking scheme would produce a watermark that is distortion-free (as defined in EXP-Edit[A] and Unbiased[B]) and also demonstrates resilience to potential text modifications, similar to the robustness of Unigram[C]. Unfortunately, to the best of our knowledge, ***no existing watermarking scheme meets both of these criteria simultaneously***.
>
> Therefore, the motivation behind our work is clear: we aim to ***achieve a better trade-off between quality and robustness by explicitly optimizing the quality and robustness objectives in an end-to-end manner***.
>
> In addition to the quality/robustness trade-off, we would like to highlight several key factors of EXP-Edit and Unbiased that compromise, in order to achieve distortion-free watermark embedding, especially when compared to our method:
>
> 1. **Detection Time Complexity**
>    We compare the average time required to detect a single watermarked sample using KGW, EXP-Edit, Unbiased, and our method. The experiment is conducted with Llama2-7B on a single NVIDIA A6000 GPU, with the following results:
>
>    | Method      | Required Time (seconds) |
>    |-------------|-------------------------|
>    | KGW         | 0.3                     |
>    | EXP-Edit    | 80                      |
>    | Unbiased    | 3.4                     |
>    | Ours        | 0.005                   |
>
>    Our method is highly efficient, requiring no access to the LLM and benefiting from GPU parallel acceleration. In contrast, Unbiased requires additional access to the LLM and prompts, while EXP-Edit has significantly longer detection times. Our method is **16,000 times** faster than EXP-Edit and **680 times** faster than Unbiased, making it feasible for scalable watermarking systems.
>
> 2. **Accessibility**
>    Unbiased requires access to the token logits of the LLM API and the prompt, which could reduce its accessibility. In contrast, our method, similar to KGW, only requires the text to be detected, making it simpler to deploy.
>
> 3. **Choice of LLM Decoding Strategy**
>    Both EXP-Edit and Unbiased work by manipulating the sampling process, and thus do not function with beam search due to the deterministic nature. As shown in Appendix I, our method performs even better with beam search than with multinomial sampling. This is because beam search tends to select higher-probability tokens, which implicitly favors more of the green-list tokens in the generated text.
>
> 4. **Low-Entropy Scenarios**
>    EXP-Edit and Unbiased are not effective in generative processes with low entropy, such as code generation, as empirically shown in the MarkLLM benchmark [E]. In contrast, our method demonstrates superior performance in code generation, as illustrated in Fig. 6 (c). Our method is logits-based and can be further enhanced with techniques like SWEET[D] to improve performance in low-entropy scenarios.
>
> We hope this clarifies the motivation behind our work and emphasizes the practical advantages of our approach. Thank you for your valuable feedback.
>
> [A] Kuditipudi et al. Robust Distortion-free Watermarks for Language Models, TMLR 2024
>
> [B] Hu et al. Unbiased watermark for large language models, ICLR 2024
>
> [C] Zhao et al. Provable Robust Watermarking for AI-Generated Text, ICLR 2024
>
> [D] Lee et al. Who Wrote this Code? Watermarking for Code Generation, ACL 2024
>
> [E] Pan et al. MarkLLM: An Open-Source Toolkit for LLM Watermarking, EMNLP 2024

---

### Official Review · Reviewer_JP2f · 2024-11-03

**Soundness:** 3
**Presentation:** 3
**Contribution:** 3
**Rating:** 6
**Confidence:** 4

**Summary:**

The authors present an end-to-end training-based text watermarking method aimed at achieving an optimal trade-off between text quality and robustness, leveraging the logits-based watermarking framework introduced by Kirchenbauer et al. Specially, they jointly train additional encoder to generate logits perturbation to shift the tokens’ probability distribution and additional decoder to extract the watermarking signals from the text. In addition, the authors introduce distortion module, address the non-differentiable operations in the end-to-end training pipeline, and consider the generalization to different LLMs.

**Strengths:**

* A distortion module is helpful to enhance the robustness.

**Weaknesses:**

*  Insufficient coverage of relevant related work
* Inadequate explanation of key methodological design choices
* Evaluation on outdated LLM architectures
* Limited adaptive attack evaluation
* Unclear figure captions (specifically Fig. 2)

**Questions:**

### **Comparative Analysis**

* The paper briefly mentions other training-based methods (UPV, SIR) but lacks detailed comparison
* Please provide in-depth analysis of architectural differences and performance variations between this work and existing training-based approaches

### **Efficiency and Generalization**
* The cross-model inference time overhead is significant - what optimizations are possible?
* How does the method handle LLMs not included in the cross-model converter?
* What is the failure mode analysis?

### **Evaluation Scope**
* Evaluation should include more recent LLMs (e.g., Yi, Qwen)
* Need broader testing across model architectures and scales

### **Related Work and Claims**

* Notable omission of generation-based watermarking methods (e.g., AWT [1], REMARK-LLM [2])
* The "first end-to-end framework" claim requires more careful qualification

[1] Adversarial watermarking transformer: Towards tracing text provenance with data hiding

[2] REMARK-LLM: A robust and efficient watermarking framework for generative large language models

### **Security Analysis**

* How does the method perform against adaptive attacks where adversaries have full access to the system?
* Need evaluation of undetectability and robustness when attackers can obtain paired watermarked/unwatermarked samples

### **Architecture Choices**
* Please justify the selection of LSTM as the decoder backbone
* What alternatives were considered and why were they rejected?

---

> ### Author Response · Authors · 2024-11-22
>
> ### Weaknesses
> - We have revised the related work section [A] and added a discussion in Appendix H, highlighting the advantages of logits-based watermarking compared to generation-based methods.
> - Our network design prioritizes efficient watermark embedding and detection.
> - While our experiments utilize SOTA open-source LLMs, we acknowledge the importance of evaluating our method on the latest architectures, which will be addressed in future work.
> - Following the recent benchmark work [B], we present attack evaluation results in Fig. 4 and Fig. 9, with detailed analysis provided.
> - The caption for Fig. 2 has been revised to enhance clarity.
>
> ### Q1: Comparative Analysis
> - We evaluate our model against training-based methods (UPV and SIR) in terms of effectiveness, robustness, and text quality. The results demonstrate how our model achieves a favorable trade-off between robustness and quality.
>
> ### Q2: Efficiency and Generalization
> - The inference overhead arises from tokenizing $k$ sequences per step. This can be mitigated through parallel tokenization, reducing time complexity by up to $1/k$.
> - As shown in Fig. 4, the similarity in tokenization results across tokenizers enables our converter to perform well if the target sentences align with the training-phase tokenizer.
> - Potential failure modes include handling sentences with unrecognized symbols or characters by the training-phase tokenizer.
>
> ### Q3: Evaluation Scope
> - While we have compared our method with several SOTA open-source LLMs, we recognize the need for broader testing across diverse architectures and scales. Due to time constraints, this will be explored in future research. We agree that this is a valuable area for further study.
>
> ### Q4: Related Work and Claims
> - Additional related works have been incorporated in Appendix H.
> - The "first end-to-end framework" refers specifically to logits-based LLM watermarking, as stated in the abstract.
>
> ### Q5: Security Analysis
> - The watermark encoder should be protected as a private key (as in KGW), while the watermark decoder can be publicly accessible due to the neural network's black-box nature.
> - Appendix E includes experiments on undetectability when attackers have access to paired watermarked/unwatermarked samples. Results show increased detectability compared to unpaired samples in Table 2.
>
> ### Q6: Architecture Choices
> - The LSTM structure mimics the KGW-based method, where green/red lists are derived from preceding tokens. The LSTM network captures temporal dependencies from preceding sequences.
> - The lightweight architecture ensures efficient watermark embedding and detection. Without resource limitations, advanced architectures like Transformers could be employed in the watermark encoder/decoder.
>
> [A] Abdelnabi, Sahar, and Mario Fritz. "Adversarial watermarking transformer: Towards tracing text provenance with data hiding." 2021 IEEE Symposium on Security and Privacy (SP), 2021
>
> [B] Leyi Pan, Aiwei Liu, Zhiwei He, Zitian Gao, Xuandong Zhao, Yijian Lu, Binglin Zhou, Shuliang Liu, Xuming Hu, Lijie Wen, et al. Markllm: An open-source toolkit for llm watermarking. arXiv preprint arXiv:2405.10051, 2024

---

> > ### Comment · Reviewer_JP2f · 2024-11-25
> > **Raise my rating**
> >
> > The additional experimental results (including SIR) in Appendix A, B, and D provide a more detailed comparison of the performance in terms of the trade-off between robustness and text quality, as well as token distribution bias, among non-training-based methods, existing training-based methods, and the authors’ method.
> >
> > Based on these efforts, I have raised my scores.
> >
> > By the way, there are additional questions you can provide some response if available.
> >
> > 1. Could you explain why the UPV's F1 Score under BERT-Large in Table 2 is 0?
> >
> > 2. Have you evaluated the watermark performance under beam search?

---

> > > ### Author Response · Authors · 2024-11-25
> > >
> > > Thank you very much for your positive feedback and for raising your scores based on our additional experimental results. We appreciate your continued engagement with our work.
> > >
> > > Regarding your additional questions:
> > >
> > > ### **The F1 Score of UPV**
> > > An F1 score of 0 indicates that all watermarked samples were misclassified. Meanwhile, an accuracy of 0.480 suggests that some non-watermarked samples were also misclassified.
> > >
> > > ### **Watermark Performance Under Beam Search**
> > > We use multinomial sampling as the LLM decoding strategy, and the detection results for beam search are further presented in Appendix I. Overall, our method continues to demonstrate superior performance and robustness.

---

### Official Review · Reviewer_QAhy · 2024-11-04

**Soundness:** 3
**Presentation:** 3
**Contribution:** 3
**Rating:** 6
**Confidence:** 3

**Summary:**

In this paper, the authors present a novel logit-based watermarking pipeline for text generation. Their approach incorporates Gumbel-Softmax for sampling and an online prompting technique for adversarial edits, allowing the encoder and decoder to be trained in an end-to-end fashion. The method achieves state-of-the-art detectability under various attacks while maintaining high-quality generated text.

**Strengths:**

- The paper includes an extensive section on experiments, including many state-of-the-art methods and attack scenarios.
- The results for overall detectability and text quality look promising.
- The encoder and decoders are small, so although an extra watermark encoder and decoder have been introduced, the generation and detection are very efficient.

**Weaknesses:**

- The result on generation diversity is not great as the proposed method has the lowest diversity among all other methods. Even though this doesn't affect the results on the benchmarks, I think this might be a bad feature for certain tasks, like synthetic data generation.
- The proposed method is training-based not like some of the baselines. The method might suffer OOD issues that the distribution of the prompt at the inference time is quite different from the training.
- The proposed method used a classifier for detection, and this does not give us an interpretable result like a p-value. This might also be bad if we want to control the false positive rate during detection.

**Questions:**

- I wonder how expensive is the training especially the requirement for the GPU memory. If I understand it correctly, the forward looks like: first token logits -> encoder -> sample the first token -> second token logits -> encoder -> sample the second token -> ... So we recursively call the encoder for n times if we generate n tokens. Would the computational graph be huge? Especially you also have to sample some tokens from the online text editor later. I wonder how did you train it exactly.
- As I mentioned above, the method might suffer OOD issues. Would the encoder/decoder trained on WikiText-103 still be effective for other datasets?
- Another thing that confuses me is that: where do you show the results for cross-llm inference? I noticed you mentioned: "To train our end-to-end model, we chose OPT-1.3B as the online LLM for efficiency." Does this mean results for llama models are the transfer inference results? Or this is for the online text editor.

---

> ### Author Response · Authors · 2024-11-22
>
> ### W1: Robustness/Quality Trade-off
> - For the explanation of the log diversity, please refer to the "Response to All Reviewers."
>
> ### W2: OOD
> - We have fully addressed the OOD problem in our evaluation. The prompts used in the training and testing phases come from different datasets. For training the watermark encoder/decoder, we use the WikiText-103 dataset, which contains articles from Wikipedia. For evaluation, we use the RealNewsLike subset of the C4 dataset, which contains news articles that differ significantly in content from the training set. The results in Fig. 5 demonstrate that our method is robust to OOD scenarios.
>
> ### W3: Detection and Interpretability
> - We use a deep classifier for detection and provide flexibility in controlling the false positive rate by adjusting the threshold for the decoder’s output logits. The ROC-AUC curve in Appendix B illustrates our model’s ability to effectively manage false positive rates through the threshold.
> - Additionally, for any given sentence, we can retrieve the logits perturbation (output of the watermark encoder) for each token as shown in Fig. 11, allowing us to determine whether tokens fall into green/red lists. This capability enables the interpretability of the watermark signal, similar to the method used in KGW.
>
> ### Q1: Watermark Embedding and Model Training
> - The watermark embedding is applied at each generation step during the training phase. Our end-to-end model was trained on a single NVIDIA RTX A6000 48GB GPU for 35k steps with approximately 5 days and the GPU memory usage is 21.96 GB. By adjusting the maximum number of generated tokens (set to 100 in our training), we can control the complexity of the computational graph. Further details of our model training are provided in Appendix F.
>
> ### Q2: Addressing OOD
> - Please refer to the response for 'W2: OOD'.
>
> ### Q3: Cross-LLM Evaluation
> - Our model is trained with OPT-1.3B and evaluated on Llama-2 and Llama-2-Chat for cross-LLM inference.

---

> ### Author Response · Authors · 2024-11-26
> **Request for Timely Feedback on our Clarification of Quality Trade-offs and OOD Concerns**
>
> Dear Reviewer QAhy,
>
> We apologize for reaching out again, but with the rebuttal discussion period concluding in **less than 36 hours**, ***Nov 26, 11:59 PM AoE***, we sincerely appreciate the continued time and effort you dedicate to discussing our submission.
>
> To demonstrate the flexibility of our model in adjusting quality metrics trade-offs, Table 5 shows that setting \( k \) to 40 enhances log diversity with a slight increase in perplexity. The ROC-AUC curves in Appendix B show that our model effectively manages false positive rates through threshold adjustments. Fig. 11 illustrates the watermark encoder's output values for each token in a sentence, identifying whether tokens fall into green or red lists, thereby enhancing the interpretability of our method. Detailed training information is provided in Appendix F, and all concerns about OOD issues in prompts and LLMs have been addressed.
>
> We are truly grateful if you could let us know if there are any remaining questions or concerns about our submission. If our responses satisfactorily address your concerns, we kindly hope you might consider reflecting this in your review score.
>
> Thank you once again for your thoughtful feedback and for your consideration.
>
> Best regards,
> Submission9685 Authors

---

> > ### Comment · Reviewer_QAhy · 2024-11-26
> > **Thanks for your response**
> >
> > I sincerely appreciate the authors' detailed response. I recognize that there were some misunderstandings on my part, which the authors have effectively clarified. Especially, the model is only trained with a small OPT-1.3B model and applied to larger Llama models. I feel like such transferability makes the method easy to use for different models and makes it kind of "training-free." Therefore, I raised my score to positive.

---

> > > ### Author Response · Authors · 2024-11-27
> > >
> > > Thank you very much for your kind and thoughtful feedback. We are delighted to hear that our clarifications were helpful. We are grateful for your positive evaluation and the time you have taken to review our work.

---

### Official Review · Reviewer_sSkQ · 2024-11-04

**Soundness:** 3
**Presentation:** 3
**Contribution:** 3
**Rating:** 6
**Confidence:** 5

**Summary:**

The authors propose an end-to-end optimized watermarking method for large language models to enable the detection of AI-generated content. The goal is to enhance the robustness/text quality trade-off of current LLM watermarking methods. The challenge is that many operations, such as generating sequences of text, are not differentiable. The authors overcome this issue by using the well-known Gumbel-Softmax trick to backpropagate through the text-generating process. To enhance robustness, the authors incorporate a paraphrasing model during the optimization method, and they develop cross-LLM adapters to train on one LLM and deploy it to other LLMs. They show robustness against six text modification attacks and improved text quality.

**Strengths:**

- The method works well and allows adapting against paraphrasing attacks during optimization.

- The authors thoroughly evaluate their approach by including experiments on robustness, detectability and impact on runtime during inference.

- The paper is clear in its presentation and presents the proposed ideas well.

- The cross-LLM inference adapter is a great idea, and I have not seen one before for trainable watermarking methods.

**Weaknesses:**

- The results from Figure 5 in their current form are not reproducible and lack transparency. I believe it should be a scatter plot that includes the quality degradation, and the authors should state the hyperparameters for each approach used for paraphrasing (e.g., the prompt used for paraphrasing).

- Abdelnabi et al. [A] have previously proposed end-to-end watermarking for LLMs. They also use the Gumbel-softmax trick to differentiate through the text generation process. The authors should consider citing this work.

- Figure 7, showing the difference in token distribution for the top 50 tokens, is difficult to interpret. It looks like the distance to the non-watermarked text is quite large (especially compared to KGW). Also, the choice of using 400 non-watermarked/watermarked samples is unclear. I think it would be better to plot detection accuracy against the size of the training dataset.

- It is well known that perplexity is an unreliable metric used to measure text quality [C]. I was surprised that the authors did not include watermarked samples in their Appendix. There is a known problem: training LLMs with Gumbel-softmax is unstable and can lead to poor results for text generation [D]. Could the authors please show watermarked samples and (potentially) include a limitation section on current challenges when using this optimization method?

--------
[A] Abdelnabi, Sahar, and Mario Fritz. "Adversarial watermarking transformer: Towards tracing text provenance with data hiding." 2021 IEEE Symposium on Security and Privacy (SP). IEEE, 2021.

[C] Wang, Yequan, et al. "Perplexity from plm is unreliable for evaluating text quality." arXiv preprint arXiv:2210.05892 (2022).

[D] Yu, Zhang Ze, et al. "Fine-tuning Language Models with Generative Adversarial Reward Modelling." arXiv preprint arXiv:2305.06176 (2023).

**Questions:**

- I do not understand why the prompt between non-watermarked and watermarked texts needs to differ (footnote 3 on page 9). Why can't the attacker re-use the same prompts when querying non-watermarked texts?

- In Figure 2, I am unclear how the authors calculate the distance $L_{sem}$ between the watermarked and non-watermarked texts $X_{wm}, X_{nwm}$. Since both sequences will differ in the sampled tokens, they will diverge throughout the generation process if sampled for many tokens. Then, calculating this semantic distance will be meaningless as you cannot effectively align $X_{wm}, X_{nwm}$. Also, it appears unreasonable that the averaged similarity over many contexts will be a meaningful measure of the overall similarity between two sequences. I would appreciate the authors elaborating on this point and providing more context

- The description of the online text editing module is a bit confusing to me. Do the authors also use Gumbel-softmax for the online text editor, or do they pass $X_wm$ directly to the decoder $D$ contrary to what is shown in Figure 1? Since the text generation process from the online text editor $N$ is not necessarily differentiable unless you use some trick, end-to-end training from the detector's prediction back to the encoder won't be possible. I would appreciate it if the authors could elaborate on this point.

---

> ### Author Response · Authors · 2024-11-22
>
> ### W1: Transparency and Robustness/Quality Trade-off
>
> - We understand your concerns regarding transparency. To address this, we have introduced robustness/quality scatter plots based on watermark strength in Fig. 8. These plots demonstrate that our model achieves the best robustness/quality trade-off overall.
> - We strictly follow the attack settings established by prior works [A] and [B]. For added clarity, we have included a table in Appendix G that provides descriptions, specific prompts, and hyperparameter settings for each attack presented in Fig. 5.
>
> ### W2: Related Work of Generation-based Methods
>
> - We have included a citation to [C] and added a discussion on the advantages of logits-based watermarking compared to generation-based methods in Appendix H.
>
> ### W3: Token distribution and undetectability
>
> - For the explanation of token distribution, please refer to the "Response to All Reviewers". There is indeed an inherent trade-off between robustness and token bias. As shown in Fig. 12, reducing the watermark strength helps lower token bias, though this adjustment could compromise robustness.
> - We present detection accuracy against training size for classifying paired (using the same prompt) watermarked/non-watermarked samples in Appendix E. The results demonstrate that existing methods (including our method) can be detected by an external classifier with paired samples when the training size is sufficient. However, we argue that such paired samples are unlikely to be accessible to adversaries, as the adversaries can only request watermarked text from the watermarked LLM API and search for unwatermarked text elsewhere. However, if watermarked/non-watermarked samples are obtained using identical prompts from different LLMs, a detector could exploit the domain gap between the two LLMs to achieve high accuracy with trivial solutions. Thus, we still conduct the paired samples experiment for reference.
>
> ### W4: Watermarked Samples and Training Pipeline
>
> - We have added examples of watermarked and non-watermarked text samples in Appendix C to validate the quality of our watermarked sentences.
> - The parameters of the online LLM are frozen during training so that the online LLM is not updated in training. Therefore, the use of Gumbel-Softmax does not degrade the quality of the LLM output.
> - Since we involve the entire LLM in the training pipeline for backpropagating gradients, our method requires relatively large computational resources and time compared to existing training-based methods. We have included detailed information about the training process in Appendix F. Despite this, once the model is trained, we develop an efficient converter for cross-LLM inference, ensuring that the computational cost during inference remains low.
>
> ### Q1: Detectability Experiment
>
> - We follow the undetectability experiment setup from [A], where the prompts for watermarked and non-watermarked texts differ.
> - Refer to ‘W3: Token distribution and undetectability’
>
> ### Q2: Embedding Domain and Semantic Similarity
>
> - The $L_{\text{sem}}$ term is not a novel contribution but a widely used method to compute semantic similarity between two sentences [D].
> - At the beginning of the generation, the strong alignment between $X_{\text{wm}}$ and $X_{\text{nwm}}$ provides solid supervision. As the sequences diverge, gradients flowing back from $L_{\text{sem}}$ at every step still help the watermark encoder learn to minimize semantic differences.
> - We argue that computing the $L_{\text{sem}}$ term across a batch of samples remains effective, as shown in the ablation study in Table 4. Removing $L_{\text{sem}}$ leads to a significant increase in perplexity.
>
> ### Q3: Gumbel-Softmax and Online Text Editor
>
> - The Gumbel-Softmax is employed in the online text editor $N$ to introduce randomness. $N$ is activated with a probability of 0.5, ensuring that the decoder also receives non-edited watermarked text $X_{\text{wm}}$.
>
> [A] Leyi Pan, Aiwei Liu, Zhiwei He, Zitian Gao, Xuandong Zhao, Yijian Lu, Binglin Zhou, Shuliang Liu, Xuming Hu, Lijie Wen, et al. Markllm: An open-source toolkit for llm watermarking. arXiv preprint arXiv:2405.10051, 2024
>
> [B] Aiwei Liu, Leyi Pan, Xuming Hu, Shiao Meng, and Lijie Wen. A semantic invariant robust watermark for large language models. In Proc. Int. Conf. Learn. Representat., 2024.
>
> [C] Abdelnabi, Sahar, and Mario Fritz. "Adversarial watermarking transformer: Towards tracing text provenance with data hiding." 2021 IEEE Symposium on Security and Privacy (SP). IEEE, 2021.
>
> [D] Sachin Chanchani and Ruihong Huang. Composition-contrastive learning for sentence embeddings. In Proceedings of the 61st Annual Meeting of the Association for Computational Linguistics, 2023.

---

> ### Author Response · Authors · 2024-11-26
> **Request for Timely Feedback on our Clarification of Robustness/Quality Trade-offs and Training Details**
>
> Dear Reviewer sSkQ,
>
> We apologize for reaching out again, but with the rebuttal discussion period concluding in **less than 36 hours**, ***Nov 26, 11:59 PM AoE***, we sincerely appreciate the continued time and effort you dedicate to discussing our submission.
>
> To ensure clarity regarding the robustness-quality trade-offs, we have provided additional explanations, including detailed comparisons in Appendix A that demonstrate the superiority of our model. Furthermore, we have included comprehensive training details in Appendix F and have addressed all the weaknesses and questions raised earlier.
>
> We are truly grateful if you could let us know if there are any remaining questions or concerns about our submission. If our responses satisfactorily address your concerns, we kindly hope you might consider reflecting this in your review score.
>
> Thank you once again for your thoughtful feedback and for your consideration.
>
> Best regards,
> Submission9685 Authors

---

> > ### Comment · Reviewer_sSkQ · 2024-11-26
> >
> > Dear Authors,
> >
> > Thank you for your response. My main concern is that I am unclear on how the paper improves over related work. The abstract states that it is about the trade-off between text quality and robustness; however, when I look at Figure 5, the robustness of the proposed method is not significantly higher than that of other methods. Similarly, for Figure 6, the quality scores appear similar to those of all other approaches. Your method is trainable, which is interesting but has, in principle, been done before (even though, as you correctly point out, Abdelnabi et al. [C] focus on seq2seq watermarking). However, your approach increases complexity as it requires a lot of computational resources to train and lacks reproducibility, as the process is much more involved than any of the other methods you compare with, such as KGW. Could you please clarify this point for me? Thank you.

---

> > > ### Author Response · Authors · 2024-11-26
> > >
> > > Thank you for your thoughtful feedback and continued engagement with our work. We truly appreciate your insights and are pleased to address your comments in detail below:
> > >
> > > ### 1. Superior Performance of Our Method
> > > We invite you to review our updated manuscript and the additional experimental results provided in ***Appendix A***. These results offer a comprehensive comparison with state-of-the-art (SOTA) competitors. Our model demonstrates significant improvements across multiple dimensions: a ***5.33% increase in robustness***, a ***9.76% reduction in perplexity (PPL)***, and a ***7.88% boost in downstream task performance***.
> > >
> > > ### 2. Difference from Generation-Based Methods
> > > Our end-to-end approach fundamentally differs from Abdelnabi et al. [C], particularly in the watermark embedding process. Abdelnabi et al. [C] embed watermarks post-generation, while our method embeds the watermark during text generation. A detailed discussion on the advantages of logit-based methods (including ours) over generation-based methods (Abdelnabi et al. [C]) is provided in Appendix H.
> > >
> > > Our contribution lies in proposing the ***first end-to-end model specifically designed for the logit-based watermarking scheme***. This approach integrates the entire LLM into the training pipeline, addressing non-differentiable modules in such a pipeline through our innovative online prompting and our model demonstrates superior performance over key metrics.
> > >
> > > ### 3. Reproducibility
> > > To ensure reproducibility, we have included our model's code in the supplementary materials. Additionally, we provide extensive details to address your concerns, including further experimental results in Appendix A, training details in Appendix F, and attack configuration details in Appendix G.
> > >
> > > We sincerely thank you for your valuable feedback and hope these clarifications address your concerns. Please do not hesitate to share any further questions or suggestions.

---

> ### Author Response · Authors · 2024-11-26
>
> ### 4. Computational Resources
> While our model requires additional resources during the training phase (details provided in Appendix F), we have developed a converter to enable ***efficient and cross-LLM*** inference. As shown in Table 3, the time and memory usage for watermark detection remains negligible. Additionally, the time overhead for watermark embedding can be mitigated through parallel tokenization, reducing the time complexity by up to $1/k$. Although this optimization is beyond the scope of our current work, it can be explored further in future research.

---

> > ### Comment · Reviewer_sSkQ · 2024-11-26
> >
> > Dear Authors,
> >
> > 1. Could you please explain the difference between Appendix A and the results in Figures 5 and 6? It appears that they are inconsistent. Are these results statistically significant?
> >
> > 2. I agree that your approach differs from AWT.
> > A minor remark: The name "generation-based" in Appendix H for methods like AWT is confusing, as the watermarking happens after generation, as you correctly describe.
> >
> > 3. Good, I am happy to see that.
> >
> > 4. As I stated in my original review, the converter is a good idea.
> >
> > The main problems I have are that the improvements do not appear to be statistically significant **and that the method incurs a higher computational cost by design. Thus, its usefulness for future work might be limited. I am willing to raise my score if the authors provide convincing arguments about why they believe that is not the case.

---

> > > ### Author Response · Authors · 2024-11-26
> > >
> > > Thank you for your positive feedback and continued engagement with our manuscript. We greatly appreciate your insights and have carefully addressed your comments below:
> > >
> > > ### 1. **Difference Between Appendix A and Fig. 5 & 6**
> > > In the initial version of our manuscript, we evaluated our method using the default settings ($\delta = 1.25, k = 20$). Thus, the results shown in Fig. 5 and 6. reflect the performance of our default model.
> > >
> > > Thanks for your comments regarding robustness/quality trade-offs, we reorganized the results into Table 5 and Fig. 9 to provide a clearer comparison by presenting robustness and quality metrics in a unified table/graph. Notably, the results of our default model remain identical to those in Fig. 5 and 6.
> > >
> > > Additionally, we introduced an alternative model configuration ($\delta = 1.25, k = 40$) to demonstrate the flexibility of our approach in adjusting quality metrics to align with user preferences. For greater transparency, we also included robustness/quality scatter plots based on watermark strength in Fig. 8.
> > >
> > > ### 2. **Clarification of the Term "Generation-Based"**
> > > We acknowledge that the term "generation-based" may be confusing, as the watermarking process occurs post-generation. In the updated manuscript, we have revised this terminology to "post-generation methods" for greater clarity in Appendix H.
> > >
> > > ### 3. **Statistically Significant Improvements in Our Method**
> > > We understand the importance of verifying the statistical significance of our model's improvements. As demonstrated in Appendix A and Fig. 5 & 6, our method ***consistently*** outperforms SOTA competitors in averaged F1 scores across three LLMs. Additionally, it achieves superior performance in translation and code generation tasks, along with lower PPL, highlighting the effectiveness of our approach.
> > >
> > > To further substantiate these results, we are conducting robustness and quality experiments with a paired t-test to validate the statistical significance of our improvements. Since these experiments are time-intensive, we will share the results as soon as they are completed.
> > >
> > > ### 4. **Justification of Higher Computational Cost**
> > > While our method incurs higher computational costs compared to existing KGW-based methods, we argue that these costs are reasonable. As detailed in Appendix F, we trained our end-to-end model only using ***one single NVIDIA RTX A6000 (48GB) GPU***. For scenarios with limited GPU memory, the batch size and maximum generated tokens can be reduced, or a smaller LLM, such as OPT-125M, can be utilized.
> > >
> > > Importantly, our model is trained offline, meaning the training process does not need to occur in real-time and can be executed on the server end. Once trained, the converter is deployed for efficient inference.
> > >
> > > We hope these clarifications address your concerns, and we are happy to provide additional details or respond to further questions. Thank you again for your valuable feedback!

---

> > > ### Author Response · Authors · 2024-11-27
> > > **Statistical Significance of Improvements**
> > >
> > > We conduct additional robustness and quality experiments using a paired t-test to validate the statistical significance of our model's improvements. The results, presented in Appendix J, demonstrate that our model achieves statistically significant improvements in both robustness and quality compared to the competitors.
> > >
> > > We appreciate your suggestion, as it helped us further substantiate the effectiveness of our method. Please do not hesitate to share any further questions or suggestions.

---

> > > > ### Author Response · Authors · 2024-11-28
> > > > **Thank You for Your Review and Feedback**
> > > >
> > > > We want to express our sincere gratitude for your thoughtful and constructive feedback on our paper. We deeply appreciate the time and effort you have dedicated to reviewing our submission, and we are especially grateful for the increased score you have awarded us.

---

### Author Response · Authors · 2024-11-22
**Response to All Reviewers**

Dear Reviewers,

We sincerely appreciate the time and effort the reviewers have dedicated to evaluating our manuscript. The concerns and feedback raised during the initial review have significantly contributed to enhancing the quality of our paper. Below, we summarize the key responses to the reviewers' suggestions and questions.

### About the Robustness/Quality Trade-off

The main concern of our method is about the lowest log diversity in Fig. 6 (b). and the relatively large bias in Fig. 7 (c).

1. **Superior Overall Performance**
We argue that the importance of different quality metrics is not identical. As shown in our experiments, our model achieves superior downstream task performance despite having relatively lower log diversity and higher token bias. This is because multiple token candidates can fit within a given context, providing flexibility without sacrificing task performance. A detailed comparison of our method with SOTA competitors is presented in Table 5, accompanied by a radar graph in Fig. 9. These results demonstrate the clear advantages of our approach, including significant improvements in robustness, perplexity, and downstream task performance. This highlights our model's ability to achieve a well-balanced performance across key metrics.

2. **User-Controllable Trade-offs**
   Our model allows users to adjust the balance of different quality metrics by modifying the watermark strength δ and the top-*k* logits tokens without retraining the model. To illustrate this, we conducted additional experiments presented in Table 5, where we adjust the value of *k* to 40 while keeping other model settings unchanged. The experiments reveal that increasing the top-*k* can enhance log diversity but could slightly increase perplexity. Similarly, as shown in Fig. 12, reducing δ helps lower token bias, though this adjustment could compromise robustness.

### About End-to-End Model Training

The main concerns regarding the end-to-end model training are differentiability and complexity.

1. **Differentiability**
   It is important to clarify that all prompts and generated text remain in the embedding domain throughout the process. In our proposed online prompting, the prompt is first converted into the embedding domain and then concatenated with X_wm/X_nwm. This ensures the entire process is differentiable, as we avoid the text-embedding transformation, which is the primary source of non-differentiability.

2. **Training Resources and Hyperparameters**
   Details about the training resources and hyperparameters are provided in Appendix F.

---

### Author Response · Authors · 2024-11-24
**Request for Further Feedback**

Dear Reviewers,

We hope this message finds you well. We are writing to follow up on the responses we provided to your valuable comments on our submission. We have already addressed all the points raised.

As the discussion deadline is approaching in a few days, we kindly request your further feedback on our responses at your earliest convenience. Your insights are crucial for us to improve our work. Thank you very much for your time and consideration.

Best regards,

Authors

---

### Author Response · Authors · 2024-12-04
**Comprehensive and Final Summary on Submission9685**

Dear Area Chairs,

We sincerely appreciate the time, effort, and valuable insights provided during the review process. It is encouraging to note that **three out of four reviewers have given positive scores** to our submission, recognizing the significance of our work. The constructive feedback has been instrumental in enhancing the quality and clarity of our paper, and we are grateful for the opportunity to improve our work.

We are pleased to report that after addressing their concerns, Reviewers **sSkQ**, **QAhy**, and **JP2f** have increased their scores. However, with the deadline approaching, we would like to kindly note that if Reviewer **JkAA** raises any further questions or concerns regarding our recent responses, we do not have the opportunity to provide additional clarification.

---

### Positive Aspects Highlighted by Reviewers

#### **1. Novelty and Contribution**
**All reviewers** acknowledge the novelty of our end-to-end model, which adapts against potential attacks during optimization, thereby enhancing robustness. Reviewers **sSkQ** and **QAhy** further highlight the innovative "convert" module that enables cross-LLM inference, improving model transferability.

#### **2. Comprehensive Experimental Validation**
Reviewers **sSkQ**, **QAhy**, and **JkAA** mention the thoroughness of our experiments, which comprehensively evaluate key aspects such as robustness, quality, undetectability, and efficiency.  Reviewer **JP2f** specifically appreciates the additional experimental results in Appendices, which provide a detailed comparison of the trade-offs between robustness and text quality.

#### **3. Presentation and Clarity**
Reviewers **sSkQ** and **JkAA** praise the well-organized and accessible structure of our paper.

---

### Concerns Raised by Reviewers and Our Responses

We have summarized the main concerns raised by reviewers in our "Response to All Reviewers." In particular, we would like to address the feedback from Reviewer **JkAA**, who expressed skepticism regarding the trade-off between quality and robustness in LLM watermarking. Reviewer **JkAA** argues that distortion-free methods eliminate this trade-off, rendering our claims about achieving a better quality/robustness balance invalid.

We respectfully disagree with this perspective and have provided detailed responses in our latest reply to Reviewer JkAA. Below is a summary of our reasoning:

1. **Distortion-Free Methods Are Not Robust:**
   - No existing distortion-free watermarking scheme achieves robustness comparable to logits-based methods like Unigram.
   - Our additional experiments demonstrate that distortion-free methods, such as Unbiased, fall short in detecting watermarked text after paraphrasing. This weak robustness has also been corroborated by recent benchmarking studies, including *MarkLLM* and *Mark my words*.

2. **Compromises in Efficiency, Accessibility, and Adaptability:**
   - Distortion-free methods sacrifice efficiency (e.g., our method is **16,000 times faster than EXP-Edit** and **680 times faster than Unbiased** in detection time).
   - Accessibility is limited, **Unbiased require access to token logits from the LLM API and the prompts**, which may not always be feasible.
   - Adaptability is hindered, as distortion-free methods are **incompatible with beam search and low-entropy scenarios**.

Our proposed method does not face these limitations, demonstrating superior efficiency, accessibility, and adaptability while maintaining robustness and quality.

---

We deeply value the reviewers' thoughtful feedback, which has strengthened our work. We hope this response clarifies our position and addresses any lingering concerns.

Thank you for your continued support.

Sincerely,
Submission9685 Authors

---

### Meta-Review · Area_Chair_3LuW · 2024-12-26

**Metareview:**

This submission is proposed a learned encoder-decoder approach for text watermarking that can be learned end to end. Learning the watermark scheme end to end in this way increases performance in a number of robustness charateristics by smaller amounts, but gives up the guarantees developed in previous work.

One point that several reviewers brought up, and I concur, is that the positioning of this paper as "the first encoder-decoder/end-to-end" learned watermark is surprising. The submission relates only to work past 2024 KGW-type analytical watermarks, missing the entire generation of watermark papers before that, based on learned encoder-decoder setups in various configurations, where Abdelnabi&Fritz, 2020 is only one of the more common examples (which even quite closely relates to the mechanics of this work, aside from still using a separate model for watermark encoding and generation, which is an older style appropriate for weaker LMs). Liu et al., 2024b and  Huo et al., 2024 are a few other example brought up by the reviewers.

Secondly, I think it is interesting to note that analytic watermarks forgo method of previous work to provide guarantees of various forms, such as provable p-values - which are highly valuable for high-stakes applications such as text forensics. This issue is also at the core of the inquiry of JkAA, who provides additional details on this discussion in relationship to analytical watermarks which explore the robustness trade-off targetted by the authors.

Overall, support from positive reviewers is marginal, and I do consider these issues significant enough that I do not recommend acceptance for now. I do think this work has merit (especially around the transfer conversion part), and I hope the authors are going to revise their manuscript based on the feedback received.

**Additional Comments On Reviewer Discussion:**

Aside from points raised above, the authors discuss a number of smaller points with reviewers, such as evaluation scope, statistical confidence in their results and model training details.

---

### Decision · Program_Chairs · 2025-01-22

Reject